# Fast and slow intraplate ruptures during the 19 October 2020 magnitude 7.6 Shumagin earthquake

Yefei Bai [1,2] ✉, Chengli Liu [3] ✉, Thorne Lay [4], Kwok Fai Cheung [5] & Yoshiki Yamazaki [5]

Strong tsunami excitation from slow rupture of shallow subduction zone faults is recognized as a key concern for tsunami hazard assessment. Three months after the 22 July 2020 magnitude 7.8 thrust earthquake struck the plate boundary below the Shumagin Islands, Alaska, a magnitude 7.6 aftershock ruptured with complex intraplate faulting. Despite the smaller size and predominantly strike-slip faulting mechanism inferred from seismic waves for the aftershock, it generated much larger tsunami waves than the mainshock. Here we show through detailed analysis of seismic, geodetic, and tsunami observations of the aftershock that the event implicated unprecedented source complexity, involving weakly tsunamigenic fast rupture of two intraplate faults located below and most likely above the plate boundary, along with induced strongly tsunamigenic slow thrust slip on a third fault near the shelf break likely striking nearly perpendicular to the trench. The thrust slip took over 5 min, giving no clear expression in seismic or geodetic observations while producing the sizeable far-field tsunami.

The largest and most tsunamigenic earthquakes around the world occur in subduction zones and usually involve thrust faulting on the plate boundary between underthrusting and overriding plates. On 22 July 2020 the Alaska subduction zone hosted the large Simeonof megathrust earthquake (Fig. 1a) with moment magnitude $M_W$ 7.8 and slip at depths from 25 to 40 km below the Shumagin Islands[1–8]. The local peak tsunami amplitude was about 30 cm, but ocean bottom pressure recordings at north Pacific deep-water (DART) stations had <1 cm tsunami amplitude, largely as a result of energy trapping in shallow water on the continental shelf and de-shoaling as the leaked waves propagated to deeper water[3,5,6]. Early aftershocks did not occur near the shallower megathrust[2,3] until the largest aftershock on 19 October 2020, with $M_W$ 7.6 (Fig. 1a). The aftershocks of that event distributed along an NNW-SSE trend located seaward of the well-constrained up-dip edge of the rupture zone of the 22 July 2020 event[5,6].

The long-period seismic moment tensor for the large aftershock indicates oblique intraplate strike-slip faulting (Fig. 1a). The routine catalog aftershock locations distribute from 5 to 40 km deep, concentrated within the Pacific plate, but straddling the megathrust fault[9], with substantial activity in the upper plate (Fig. 1b). Relative relocations are required to better resolve the upper plate aftershock locations. Typically, large intraplate ruptures seaward of large megathrust events involve normal-faulting below the outer trench slope, with faults striking parallel to the trench[10,11], so the unusual 50° eastward dipping, strike-slip faulting in the October aftershock suggests a distinctive stress state in the plates along the Shumagin region. Lateral gradients in megathrust coupling[5,8,12–14] from the strongly coupled Semidi region in the northeast along the adjacent $M_W$ 8.2 1938 Alaska and 2021 Chignik interplate earthquake rupture zones to the weakly coupled Shumagin Islands region (Fig. 1a) may cause internal shearing

[1]Ocean College, Zhejiang University, Zhoushan, Zhejiang, China. [2]Hainan Institute, Zhejiang University, Sanya, Hainan, China. [3]School of Geophysics and Geomatics, China University of Geosciences, Wuhan, Hubei, China. [4]Department of Earth and Planetary Sciences, University of California Santa Cruz, Santa Cruz, CA, USA. [5]Department of Ocean and Resources Engineering, University of Hawaii at Manoa, Honolulu, HI, USA. ✉e-mail: yfbai@zju.edu.cn; liuchengli@cug.edu.cn

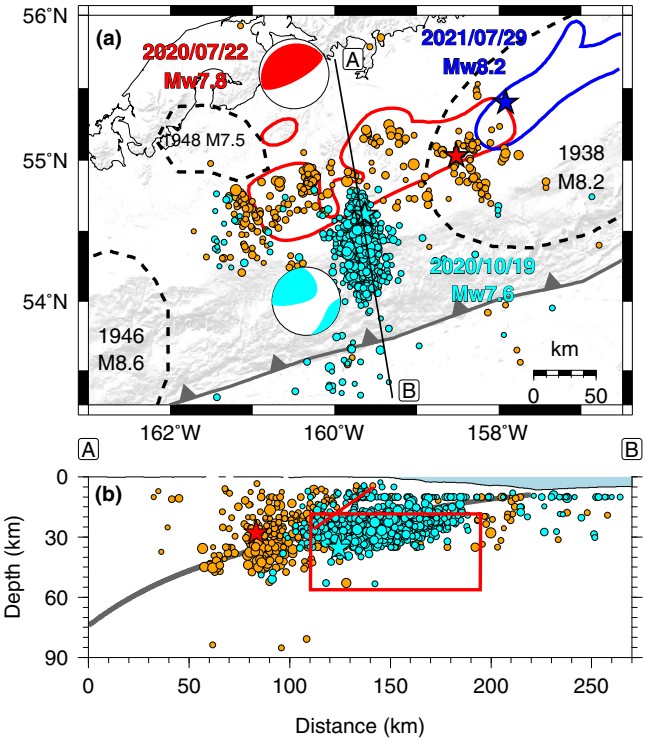

**Fig. 1 | Regional setting for the 19 October 2020 earthquake. a** Map of the 22 July 2020 $M_W$ 7.8 Simeonof megathrust event epicenter (red star), aftershocks prior to 19 October 2020 (gold circles with radii scaled by magnitude), finite-slip (>0.5 m) region[6] (red contours), and long-period moment tensor (red); the 19 October 2020 $M_W$ 7.6 aftershock epicenter (cyan star), aftershocks (cyan circles with radii scaled by magnitude), and long-period moment tensor from the USGS-NEIC (cyan); the 29 July 2021 $M_W$ 8.2 Chignik megathrust event finite-slip region (>0.5 m)[16] (blue contour) and epicenter (blue star); and nearby large historic event rupture zones (black dashed lines). Line AB indicates the position of the cross-section in (**b**), which shows the depth distribution of the aftershock sequences, position of the slab interface[9] (bold line), and projections of the two-faults in the fast-slip rupture model in Fig. 3 (red lines).

within the Pacific plate, possibly accounting for the strike-slip source mechanism[15]. The specific eastward dipping fault geometry is not an obvious outcome, but may represent reactivation of a pre-existing fault within the Pacific plate. However, there is much uncertainty in the coupling in the shallow part of the megathrust along both the Shumagin segment and the adjacent Semidi segment. There are no inter-plate thrust faulting aftershocks in the shallow megathrust for the $M_W$ 7.8 mainshock and no resolution of any shallow megathrust afterslip[8], so it is unclear what the state of coupling is for the plate boundary near the 19 October 2020 aftershock[5,8].

Adding to the unusual attributes of the 19 October 2020 after-shock are much larger observed tsunami signals at DART stations and tide gauge stations at Sand Point, Alaska and in Hawaii relative to the 22 July 2020 thrust event (Fig. 2). This is surprising because the oblique strike-slip mechanism is intrinsically less efficient in generating tsunami as it produces less vertical seafloor deformation than a thrust event, and the $M_W$ is lower than for the mainshock. Tsunami excitation is expected to increase if seafloor deformation extends seaward of the shelf break[5,6,16]; however, this tendency does not overcome the effect of the unfavorable faulting geometry. Indeed, the Pacific Tsunami Warning Center underestimated the tsunami amplitudes expected in Hawaii relative to the mainshock, which had produced very small tsunami signals observed around the islands. Hawaii was in the clear prior to the aftershock tsunami arrival, but a last-minute statewide advisory for hazardous coastal conditions was activated after signals were detected at the Hilo and Kahului tide gauges.

In this work we examine seismic, geodetic, and tsunami data for the 19 October 2020 earthquake to discover the source of the unexpected tsunami amplitude. This analysis indicates that the event has an unprecedented complex source with ruptures on either side of the plate boundary and a slow faulting process in the upper plate that generated the unexpected strong tsunami.

## Results and discussion
### Fast faulting component
Guided by the long-period moment tensor solution for the 19 October 2020 earthquake and the aftershock distribution from the U.S. Geological Survey National Earthquake Information Center (USGS-NEIC) (Fig. 1), a model of the space-time slip distribution of the rupture was developed. This was based on the inversion of teleseismic *P* and *SH* waveforms, regional broadband and strong-motion three-component recordings, and regional GNSS high-rate time series and static offsets. A planar fault with strike 350° and dip 50° (eastward) was adopted from the long-period best double couple solution, and the length and width of the model were adjusted to achieve a good fit to the data. A single fault inversion places patches of large-slip about 30 km south of the hypocenter within the Pacific plate and shallow slip north of the hypocenter, with the latter locating above the ~20 km depth of the expected megathrust boundary, similar to a USGS-NEIC model (https://earthquake.usgs.gov/earthquakes/eventpage/us6000c9hg/finite-fault). Constraining the slip to intraslab depths >20 km degraded the fit to the data, and even if allowed to cross the megathrust, the single-plane model does not account for the significant non-double couple component of the moment tensor (Fig. 1a).

A discrete second fault was introduced to account for the extra tensional component of the moment tensor. We explored many positions and orientations for faulting in the upper plate and in the subducting slab (Supplementary Fig. 1), finding that it must locate below the shelf close to the shelf break, but the depth is not well constrained because the seismic moment is low and the moment release is during or after the peak moment release of the strike-slip faulting. Assuming a northward-dipping normal fault in the upper plate below the continental shelf (Fig. 3; Supplementary Fig. 1a) as has been imaged by reflection seismology[17] allows the local GNSS and regional data to be fitted comparably well to models with northward-dipping normal faulting in the slab (Supplementary Figs. 1d, 2b,c) or southward-dipping oblique normal faulting in the wedge (Fig. 3a, Supplementary Figs. 1g, 2e,f) or in the slab (not shown). There are only minor effects on the inverted intraslab strike-slip fault slip among the models with different positions and orientations of the secondary faulting, and the secondary faulting geometries all produce non-double couple radiation patterns that match the data (Supplementary Fig. 1). However, the shallow northward-dipping option is preferred because a corresponding normal fault in the wedge has been previously directly imaged, whereas there is no indication of a shallow southward-dipping fault in the reflection images. The northward-dipping and southward-dipping faulting orientations in the slab both involve faults that intersect the main intraplate strike-slip fault, with slip extending on either side of that fault (Supplementary Fig. 2), which is considered to be very unlikely. While we prefer the shallow northward-dipping option for the secondary faulting, this choice has negligible impact on the tsunami modeling to follow.

Figure 1b shows the geometry of the preferred two-fault finite-slip model relative to the aftershocks. The rupture of the two-faults completes within 40 s, at typical 2–3 km/s rupture speed. Details of the slip distributions on the faults are shown in Supplementary Figs. 1a–c and the seafloor motions are shown in Fig. 3b. The seismic moment of the upper plate rupture is $M_O = 0.29 \times 10^{20}$ Nm ($M_W$ 7.0), much smaller than that of the intraslab rupture, $M_O = 2.5 \times 10^{20}$ Nm ($M_W$ 7.5). The model successfully predicts the full suite of seismic and geodetic

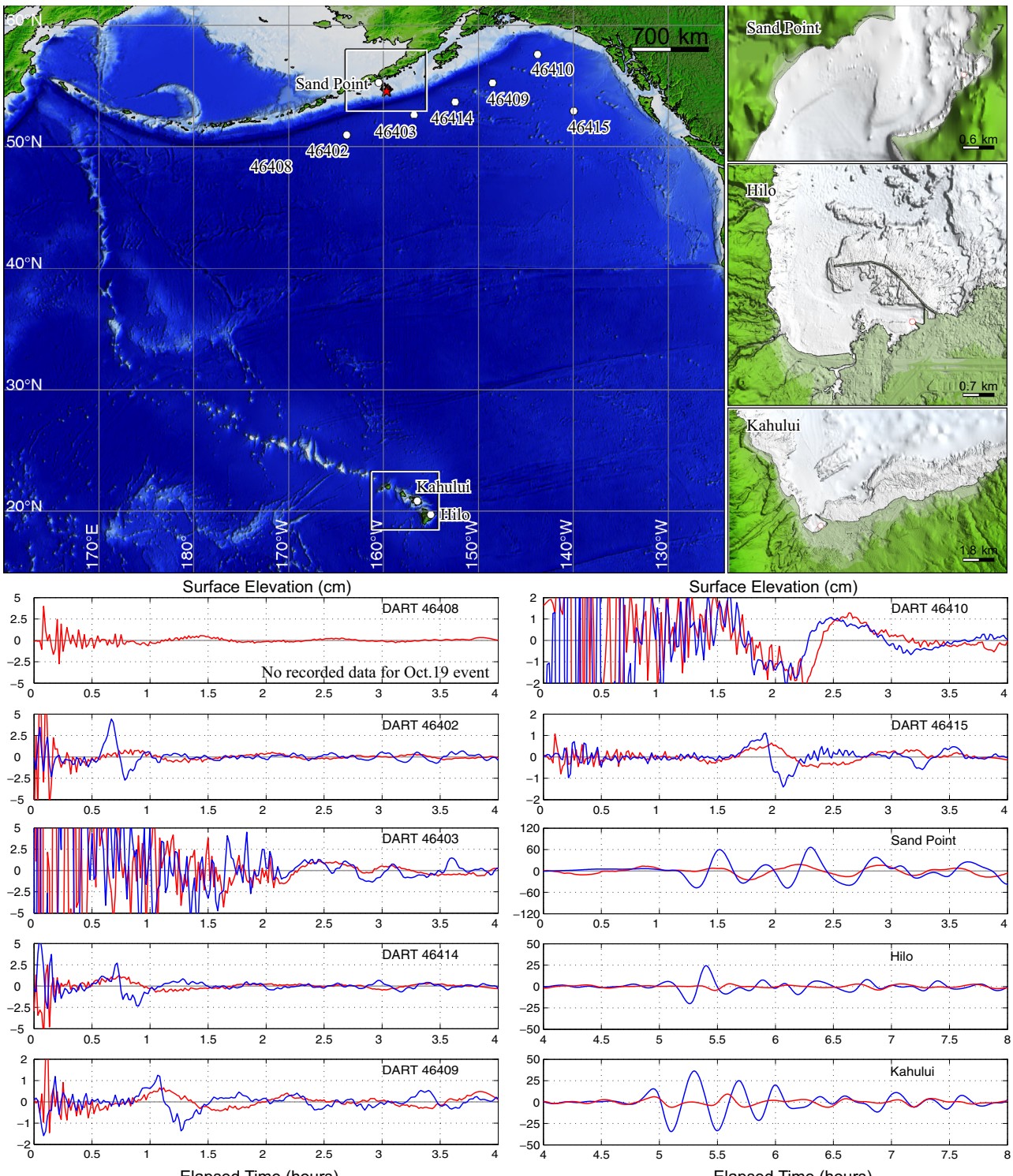

**Fig. 2 | Tsunami observations for the Shumagin Segment earthquakes.** Top left: map of north Pacific bathymetry with location of the DART stations and tide gauges at Sand Point, Alaska, and Hilo and Kahului, Hawaii (labeled white circles). The red star indicates the location of the 19 October 2020 rupture. Top right: high-resolution bathymetry near the tide gauge stations. Lower Panels: Comparison of DART, Sand Point, and Hawaii tide gauge recordings for the 22 July 2020, $M_W$ 7.8 Simeonof mainshock (red) and the 19 October 2020 $M_W$ 7.6 aftershock (blue).

observations (Supplementary Figs. 3, 4, 5, 6), including the non-double couple radiation found for long-period seismic waves. The fits to the vertical and horizontal static motions (Fig. 3a) and time histories (Fig. 3c) at the local GNSS stations AC12 AC28 are very good. The precise geometry and location of the upper plate fault is not uniquely determined, as noted above, but it cannot shift north of the intraslab

rupture, and it locates in a region of high upper plate aftershock activity; event relocations and aftershock focal mechanism studies may better constrain the exact position. It is, to our knowledge, unprecedented to detect coseismic rupture of two faults on either side of a megathrust. The occurrence of complex faulting in the upper and lower plates may be associated with the along strike gradients in

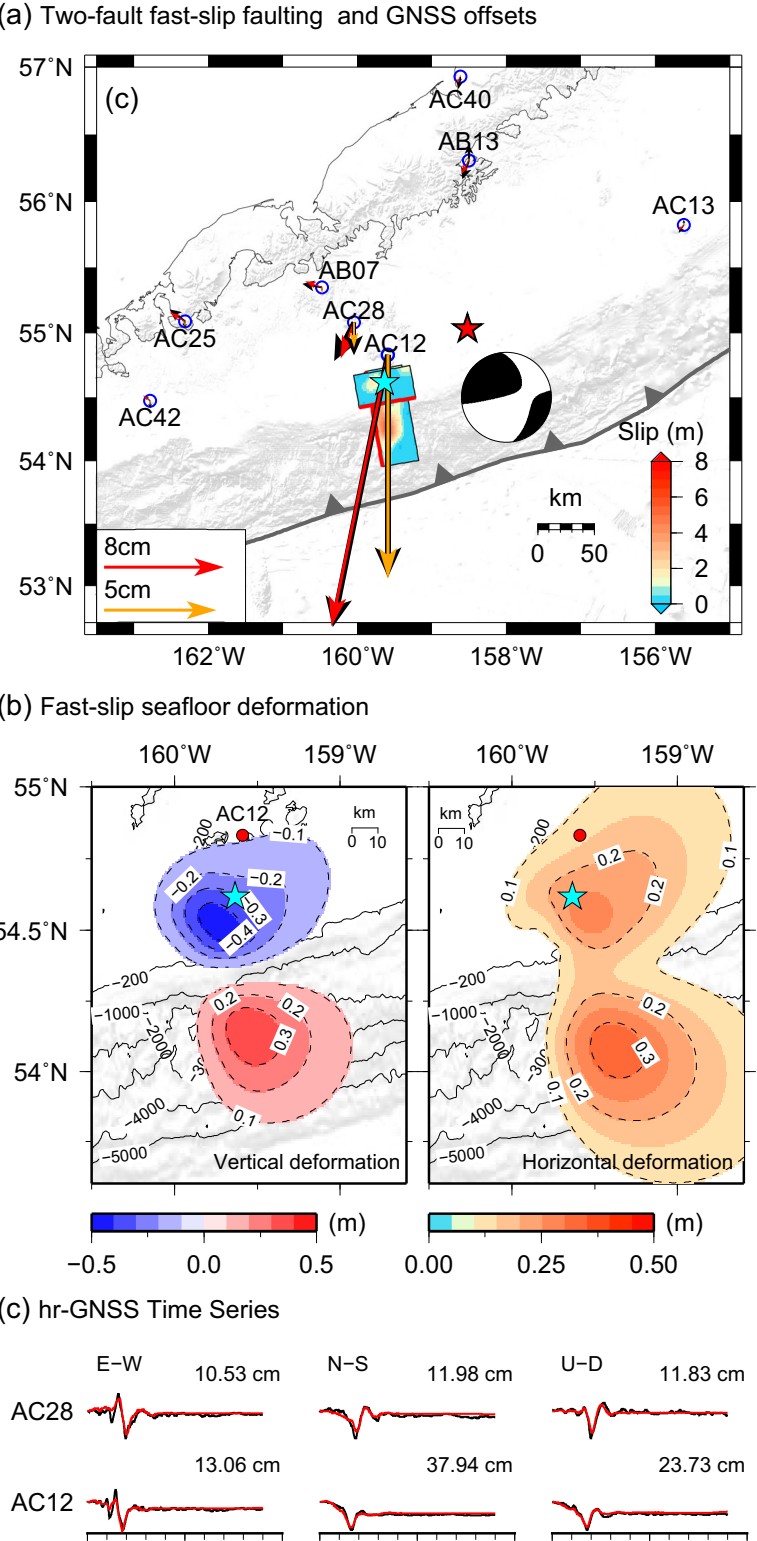

**Fig. 3 | The two-fault fast-slip finite-fault model. a** The slip distribution for two-faults is inverted from GNSS, regional broadband and strong-motion, and tele-seismic observations. The composite moment tensor for the two-fault model (black) is shown and matches the long-period solution in Fig. 1. Details of the slip distribution are shown in Supplementary Fig. 2c. GNSS coseismic offset observations (black arrows) are compared with predictions (horizontal motions in red,

vertical motions in gold). Predictions of the seismic waves and high-rate GNSS waveforms are shown in Supplementary Figs. 3, 4, 5 and 6. Cyan star denotes the epicenter, and open blue circles indicate locations of nearby GNSS stations. **b** Vertical and horizontal seafloor deformation calculated for the two-fault fast-slip model. Red circles denote the GNSS station AC12. **c** Observed (black) and computed (red) GNSS ground motions for stations AC28 and AC12.

megathrust coupling[15], but the specific coupling state of the shallow megathrust is not known with any confidence.

## Tsunami prediction

Usually, such finite-fault models produce good fits to observed tsunami waveforms, with only minor adjustments[6,16,18,19]. We model the tsunami observations using the non-hydrostatic code NEOWAVE[20,21] with excitation from the seafloor motions (Fig. 3b) for the preferred two-fault model. The model region extends across the North Pacific Ocean with increasing resolution in nested grids around the tide gauges at Sand Point and King Cove, Alaska and Kahului and Hilo, Hawaii (Supplementary Fig. 7). Despite the consistency with the seismic and geodetic observations, the two-fault fast-slip model in Fig. 3a rather dramatically fails to predict the recorded tsunami waveforms (Fig. 4). The initial DART arrivals show a broad pulse with a pointed peak or double peaks evident of superposition of two dominant harmonics. The two-fault fast-slip model reproduces the timing, initial rise, and amplitude for the first shoulder or peak of the pulse, but fails to account for the large main or second peak (positive peaks at DARTs 46402, 46414, 46409, 46415) as well as the following sea surface drawdown (negative troughs at DARTs 46402, 46414, 46409, and 46415). The computed waveform also underestimates the wave amplitude at Sand Point with an unexplained 14 min early arrival and at King Cove without time shift. The comparison at Hilo and Kahului shows little resemblance between the computed and recorded waveforms even after 4 min and 2 min shifts of the computed waveforms are made to match the recorded arrival times. There is a consistent underestimation of spectral energy over 20 min periods at most of the stations. These failings point to a missing component in the tsunami source.

## Constraining the second tsunami source

To match the observed tsunami waveforms, an additional stronger source of tsunami excitation is required, but the two-fault fast-slip model alone already adequately accounts for the full set of seismic and geodetic data. This holds even for 256 s period Rayleigh and Love waves from global stations, for which the two-fault model predicts the four-lobed radiation patterns well (Supplementary Fig. 6). From the DART waveform comparisons, the additional source must have a 4−5 min delay relative to the initial compound faulting to account for the larger second peak, yet the nearby geodetic ground motions show no deformation after the first 60 s. The earlier deformation is well accounted for by the two-fault fast-slip model (Fig. 3c). Because the tsunami wave period is inversely proportional to the square root of the source water depth, the excitation most likely includes uplift of the sea surface over the continental slope to account for the impulsive peak along with some drawdown near the shelf break to match the wide trough that follows immediately.

Given the lack of a priori information about the location of the additional source of the tsunami, we initially explored a simplified parameterization appropriate to the first order for a slump or a shallow dip-slip fault, given by a surface dipole with a two-lobed pattern of seafloor up-lift and down-drop[22]. This flexible parameterization allows adjustment of the spatial extent of the deformation zone, which along with the local water depth, influences the period of the tsunami. The absolute position, amplitude, and timing of the dipole deformation trade-off, and these parameters are systematically explored over a region seaward of the hypocenter straddled across the shelf break. This procedure defines a parsimonious representation of the second tsunami source with a clear indication of the required spatial and temporal seafloor deformation. Reverse time migration of the tsunami signals can also be attempted, but the effective source time that must be assumed to form an image of the initial sea surface displacement is uncertain. There is also the large (14 min)

time discrepancy for the one station to the north (Sand Point tide gauge), so the efficient forward modeling approach was preferred. Trial and error fitting of the signals defines a fairly narrow range of position, amplitude and timing for the required seafloor motion that can match the DART data using a simple dipole model along with the fast two-fault model.

The dipole source is a parametric function that was introduced for approximation of the seafloor deformation resulting from submarine slumps[22]. The seafloor deformation consists of a depressed region and an uplifted region aligned in the direction of steepest slope. The vertical seafloor displacement in the depressed seafloor region is defined by

$$\eta(x,y) = \eta_0 \left( \operatorname{sec} h \frac{4x}{w_x} \right)^2 \left( \operatorname{sec} h \frac{4y}{w_y} \right)^2 \tag{1}$$

where $(x, y)$ are Cartesian coordinates, $\eta_o$ is the depth, and $(w_x, w_y)$ are the nominal length and width. The uplifted region has the same geometry but with $(w_x, w_y)$ increased by a factor of $\alpha$ and $\eta_o$ decreased by $\alpha^2$ to account for run-out effects of any slump material while conserving volume[22]. The factor $\alpha$ should depend on the granular flow mechanics if this model is being interpreted to represent a submarine slump or landslide and a value of 1.21 is adopted for the present study[22], but our goal is only to guide us to model the second source with a more quantitative dislocation model given the lack of direct observational constraint.

The two-fault fast-slip model of the $M_W$ 7.6 aftershock adequately accounts for the initial arrival recorded at the DART stations (Fig. 4), as noted above. We utilize the dipole source representation as a tool to search for an additional tsunami excitation that can match the second arrival and the following trough. The timing of the dipole source and the location, orientation, depth, and dimensions of the trough are free parameters in this search. The DART records, which cover a wide azimuth of the tsunami waves seaward from the source (Fig. 2), are quite effective in constraining these parameters. In particular, the phase of the computed tsunami waveforms is very sensitive to the position of the seafloor uplift (controlled by the placement of the dipole) and its location relative to the shelf break. The onset timing of the dipole seafloor motions strictly depends on the time lag between the first two peaks shown in records of DART46414, 46409, and 46415 (Fig. 4). The position of the seafloor motion along the bathymetry contour is sensitively determined by finding consistent and accurate predicted arrival time of the largest sea surface peak at each DART. Meanwhile, the sea floor deformation location across the bathymetry contour significantly affects the wave period and detailed waveform features. Of 160 realizations, the preferred dipole model has a depressed seafloor 25 km long, 16.7 km wide, and 2 m deep and is located 0.2 arc-degree west and south of the earthquake epicenter 4 min after the faulting. The pattern of seafloor motion has about 20 km absolute position uncertainty along the shelf break, with peak seafloor deflections of ~±2 m over regions ~20 km in dimension.

The preferred dipole distribution of seafloor motion is shown in Fig. 5, alone and in combination with the motion from the two-fault solution. The tsunami predictions for the two-fault plus dipole source, with the dipole being delayed by 4 min relative to the faulting, are shown in Fig. 6. These provide good overall agreement with the tsunami observations, much improved relative to those in Fig. 4. Being a simplified representation of a slump or shallow dip-slip faulting, the solution in Fig. 5 is not intended to be a physical model of the process; it is a parameterization that captures the basic kinematics of the tsunami excitation and interference with the wave motions from the fast-slip component that can guide us in physical source modeling. Lacking detailed bathymetric information pre- and post-earthquake, it is

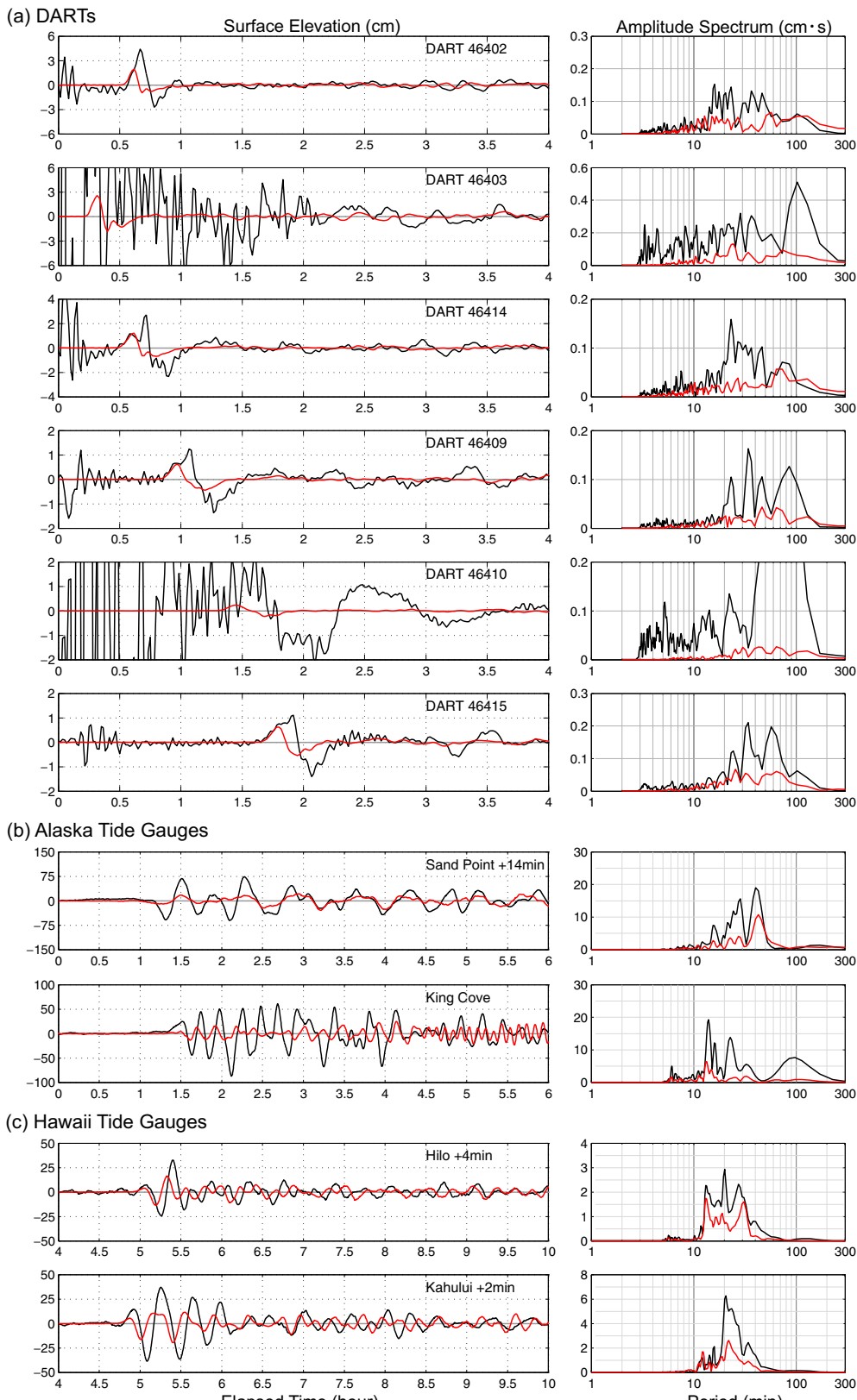

**Fig. 4 | Tsunami predictions for the two-fault fast-slip model.** Observed (black lines) and predicted (red lines) tsunami surface elevation time series (left column) and spectra (right column) for the two-fault model in Fig. 3 and Supplementary

Fig. 2c. **a** DART stations. **b** Alaska tide gauges. **c** Hawaii tide gauges (Fig. 2. and Supplementary Fig. 7). The computed time series at the tide gauges have been shifted by the indicated time to align with the recorded arrivals.

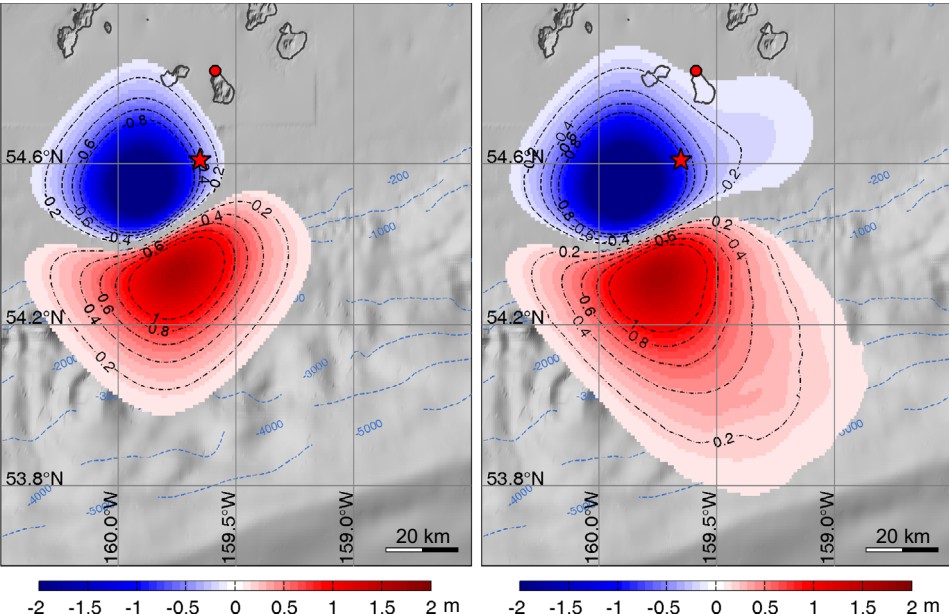

**Fig. 5 | Seafloor deformation for the dipole models.** Seafloor deformation for an optimized dipole model alone (left) and combined with the two-fault fast-slip model in Fig. 3 (right). Tsunami waveform predictions for this model are shown in

Fig. 6. Red stars indicate the epicenter, and red circles denote the GNSS station AC12.

ill-constrained to try to calculate a landslide volume or equivalent force to interpret the inferred seafloor deformation, so we do not pursue that at this time. Despite its ability to generally match the tsunami data, the non-physical dipole model does not provide predictions of horizontal motions or time history of the source producing the seafloor deformation. We use the dipole result only to guide location of candidate faulting models, with the orientation constrained to make sense relative to possible slump geometry. However, the dipole modeling is not an all-inclusive approach for evaluating seismic excitation. One can represent a slump with a shallow normal-faulting geometry, but the key constraint is that the slump must go down-slope. As shown for models below with physical arc-perpendicular dislocations, this geometry violates the hr-GNSS observations at nearby stations. Rotating the dipole (or normal-faulting) to reduce the deformation at the hr-GNSS stations violates the down-slope slumping requirement, reducing the viability of any slump interpretation in the first place.

**Unsuccessful slow-slip faulting geometries**
Given the guidance provided by the simple dipole modeling, we considered physical fault dislocation models for plausible geometries that can match the salient features of seafloor deformation from the dipole model that leads to successful match of the tsunami waveforms. This includes simultaneous assessment of the seismic and geodetic motions produced by such models for the sensitive high-rate GNSS recordings at nearby stations AC12 and AC28. The latter constraint is very important; there is essentially no geodetic or seismic signature of the second (dominant) tsunami source, and models that violate this can be rejected with confidence. We considered appropriately placed models with delayed slow thrust slip on the shallow megathrust (Methods, Supplementary Figs. 8, 9) or slow thrust slip on an upper plate splay fault with a strike parallel to the trench (Methods, Supplementary Figs. 10, 11) and allowed sufficiently long source process times to obscure the seismic and geodetic expressions while giving strong tsunami excitation, finding models that match the tsunami signals by extensive searches over model parameters (fault dimensions, slip, absolute location, etc.). However, those models that do match the tsunami observations acceptably all badly violate the geodetic observations at AC12 and AC28

(Supplementary Figs. 8, 10). This eliminates the more obvious candidate model geometries.

**Successful slow-slip faulting geometry**
In our exploration of splay fault models, we found that steepening the fault dip reduced motions at AC12 and AC28, but still violated the geodetic observations. Rotating the upper plate thrusting to be on a fault almost perpendicular to the trench reduced the predicted motions to be negligible. This gives our preferred model for the 19 October 2020 rupture, involving a combination of the two-fault fast-slip intraplate rupture in Fig. 7a and slow rupture (>5 min long, beginning 30 s after the initiation of the fast-slip) with a seismic moment of $1.8 \times 10^{20}$ Nm on a third fault dipping 30° westward with strike of 190° located below the continental slope (Fig. 7b). The combined fast- and slow-faulting model produces excellent predictions of the tsunami waveforms at DART and tide gauge stations (Fig. 8), without violating seismic or geodetic observations already accounted for by the fast faulting or producing observable deformation from 30 to 330 s during the slow-slip rupture, as indicated by the fits to AC12 and AC28 in Fig. 7d. The strike of the slow-slip faulting is constrained by the need to have seafloor drawdown at and inland of the continental shelf break to match the wide trough immediately after the initial arrivals (if too far inland the wave would get trapped by the shelf), and is resolved within ~±15°, and there is comparable uncertainty in the dip. The waveform fitting for the seismic and geodetic observations for this model is indistinguishable from that for the fast two-fault model shown in Supplementary Figs. 3–6. This extends to the 256 s spectral amplitudes, for which the long-source duration weakens the signal amplitudes to a level comparable to the secondary fast-slip faulting (Supplementary Fig. 6), but with different azimuthal pattern and a large phase shift, making it very difficult to detect. This model represents a solution with existence, but not with uniqueness. The data are all well-fit with the same slow-slip faulting using alternate choice of the second fast-slip faulting, for north-dipping intraslab (Supplementary Figs. 12, 13) or shallow south-dipping (Supplementary Figs. 14, 15) geometries. Recognizing that we are reconciling the strong tsunami excitation with no clear seismic or geodetic expression of the slow-slip component of the compound faulting, non-uniqueness is expected.

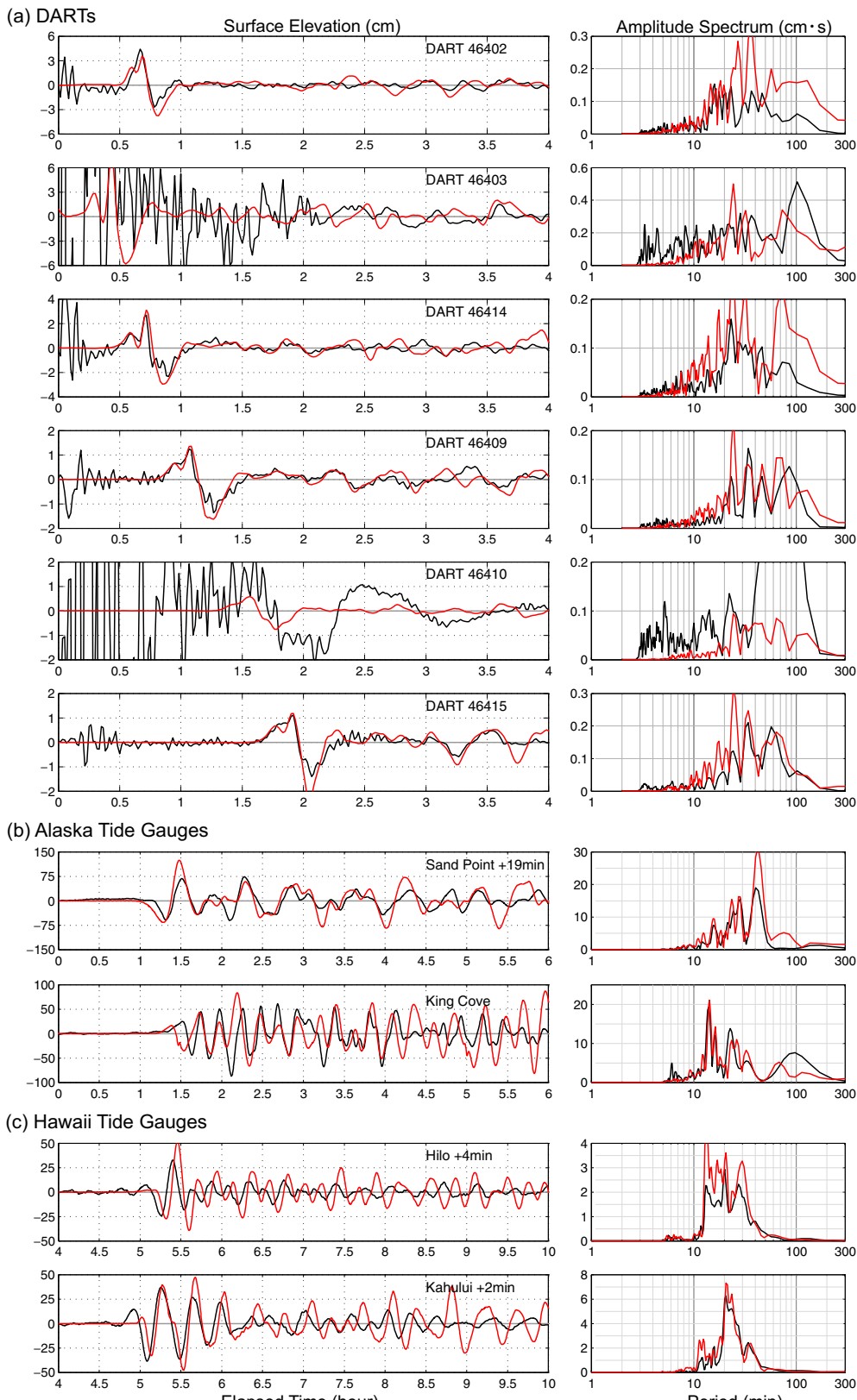

**Fig. 6 | Tsunami predictions for a two-fault plus dipole model.** Observed (black lines) and predicted (red lines) tsunami surface elevation time series (left column) and spectra (right column) for the 2-fault model plus dipole model in Fig. 5. **a** DART stations. **b** Alaska tide gauges. **c** Hawaii tide gauges. The computed time series at the tide gauges have been shifted by the time indicated to align with the recorded arrivals.

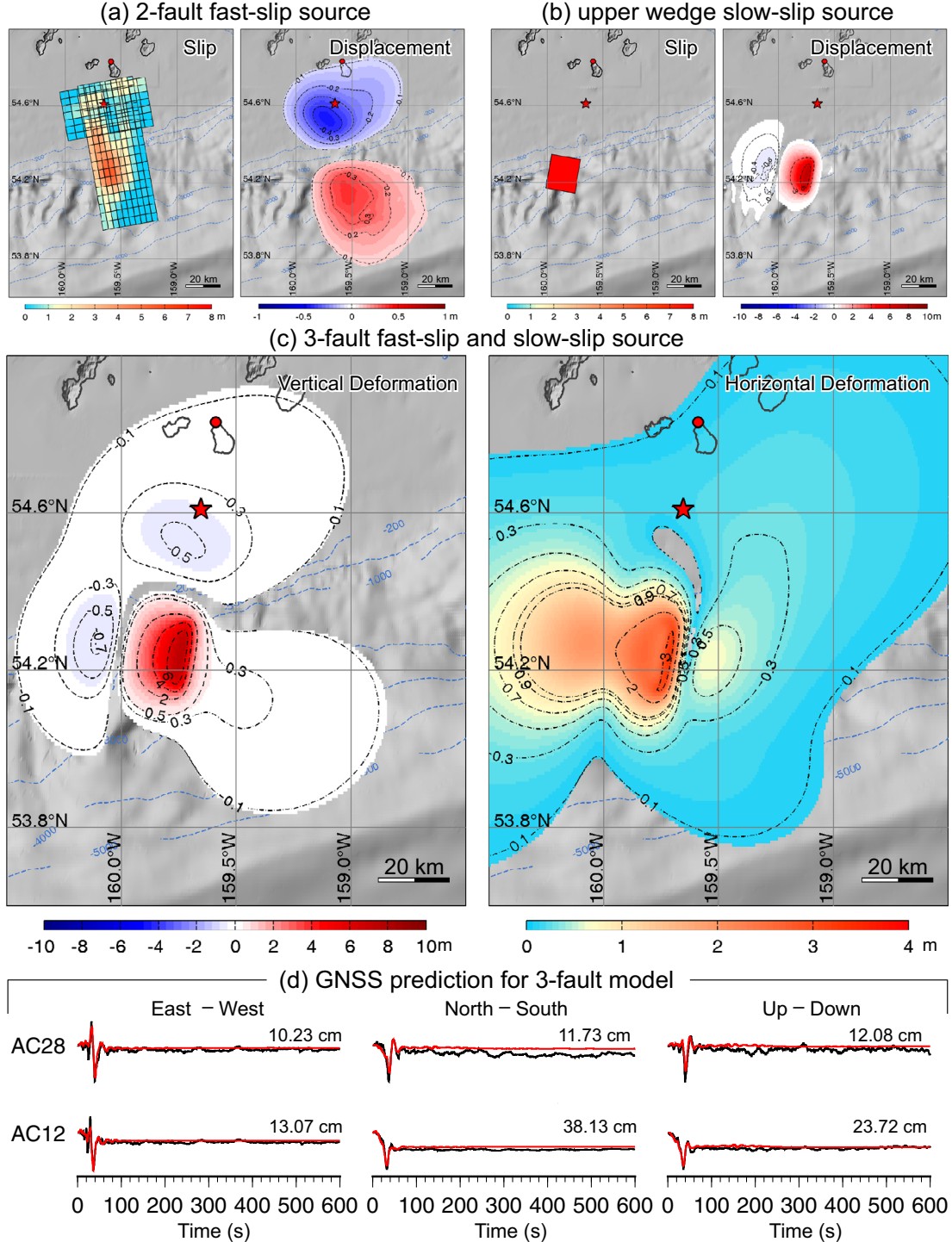

**Fig. 7 | Seafloor deformation for the two-fault fast-slip and three-fault combined fast- and slow-slip model with fits to adjacent GNSS ground motion.**
**a** Fault model slip distributions and corresponding sea floor deformation distributions for the two-fault fast-slip model in Fig. 3a. **b** An additional slow upper plate slow-slip thrust fault. **c** The superimposed total vertical and horizontal seafloor deformation from the combined 3-fault model. **d** Observed (black) and computed (red) GNSS ground motions for stations AC28 and AC12 extending over a 600 s time scale. Tsunami predictions for this model are shown in Fig. 8. Red star indicates the epicenter, and the red circle denotes the GNSS station AC12.

The specific geometry of the inferred slow thrust faulting, with along-trench compression in the upper plate, is surprising, and if this model is correct, it comprises an unexpected tsunami hazard in the region. The presence of weak sediments near the shelf break may have influenced slow-slip rupture with 15 m of slip over ~300 s, as found for this successful model, which has fault dimensions of 20 km × 20 km. Such large slip over localized area has been observed in shallow megathrusts environments, typically involving a tsunami earthquake[23] or aseismic transient slip[24]. Transpressional environments have been observed to have large slow thrust faulting along with dominant strike-slip faulting as well[25]. Models with a larger fault area (30 km × 30 km; 40 km × 40 km) and lower slip (7 m, 4 m) that have similar total moment may be viable, but it is challenging to fit all of the tsunami data as well as in Fig. 8 (e.g., Supplementary Figs. 16, 17). While lower slip is

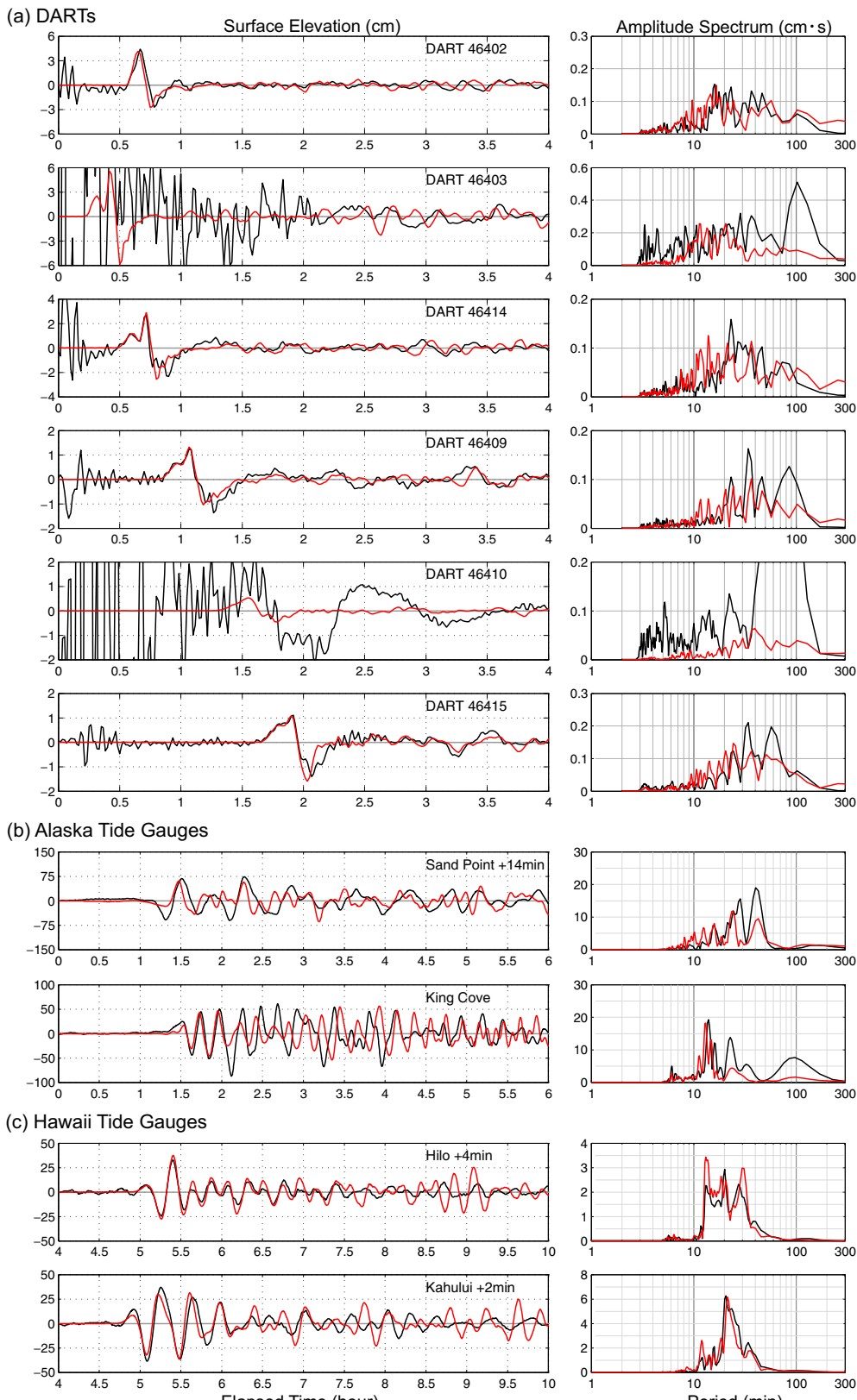

**Fig. 8 | Tsunami waveform predictions for the three-fault fast-slip and slow-slip model.** Observed (black) and predicted 3-fault (red) tsunami surface elevation time series (left column) and spectra (right column) for DART and tide gauge stations for the model in Fig. 7. **a** DART stations. **b** Alaska tide gauges. **c** Hawaii tide gauges. The computed time series at the tide gauges have been shifted by the indicated time to align with the recorded arrivals.

appealing, larger fault dimensions imply more observable faulting in the wedge, for which available bathymetry and reflection profiling now provide independent evidence. The non-unique modeling suggests slow slip of from 4 to 15 m on the westward-dipping upper plate thrust fault.

Several factors contributed to the unexpectedly large tsunami observed in Hawaii from the 19 October $M_W$ 7.6 aftershock. The fast-slip and slow-slip ruptures produce prominent uplifts on the continental slope that generates 15–45 min period tsunami waves (Fig. 8). Although the uplift and subsidence patches from the two components overlap, the spatial offset, 30 s onset delay and 300 s process time of the slow component results in phase lags and destructive interference between the two wave systems with exception of around the Shumagin Islands, where the arrivals from the two systems are aligned (Supplementary Movie 1). Supplementary Fig. 18 shows maps of the maximum wave amplitudes from the 2-fault, slow-slip, and 3-fault models, demonstrating some of the interference effects, and Supplementary Fig. 19 shows the contributions to the specific DART and tide gauge signals indicating the phase lag and interference of the fast-slip and slow-slip contributions to the waveforms. The modeled 19 October 2020 tsunami radiation wave pattern extending across the ocean contrasts strongly with the $M_W$ 7.8 mainshock event (Fig. 9), accounting for the differences seen in Fig. 2. The slow-slip near the continental break produces much larger tsunami waves across the northern Pacific Basin relative to the stronger megathrust rupture from the mainshock beneath the shelf. Importantly, the wave periods of 15–45 min are within the resonance range along the Hawaiian Islands, leading to amplification over the interconnected insular shelves[26].

The fast-slip faulting likely drove the slow-slip faulting by either dynamic or static stress changes. We evaluate the latter by computing the Coulomb stress change on the target thrust fault geometry used for the successful slow-slip component (methods and Supplementary Fig. 20). To address the uncertainty in the upper plate fast-faulting component, we considered the Coulomb stress change for both northward-dipping and southward-dipping geometries of the shallow faulting, and in both cases the specific region where the slow-slip occurred experiences modest increases in Coulomb stress of up to 0.5 MPa over the depth range 3–13 km, supporting the possibility of static triggering, and dynamic wave stresses will be even larger in the same region. While the cause of lateral compressional strain along the continental slope is unclear, it could be a manifestation of lateral variations in interplate coupling or of topography on the underthrusting plate. High-resolution 3D imaging of the source region may shed light on this question.

## Compound intraplate faulting and tsunami hazard

The extraordinary mix of fast-slip and slow-slip intraplate faulting that occurred in the 19 October 2020 $M_W$ 7.6 earthquake is summarized in Fig. 10. This full complexity is resolvable only by combining seismic, geodetic, and tsunami observations. The event is unprecedented in involving large slips on faults likely both above and below the main plate boundary and in having a mix of fast-slip and slow-slip faulting occurring on distinct faults, two of which strike near-perpendicular to the trench. Fast-slip rupture within the slab triggering coseismic slip on the north-dipping normal fault imaged in reflection profiles[17] is unprecedented, but certainly viable. The upper plate in this region involves a series of accreted terranes (notably the Prince William and Chugach Terranes) separated by north-dipping sutures[27,28].

Confirming aspects of the proposed slow-slip model will be challenging; the slow-slip faulting has large slip of 4–15 m, but may be surrounded by low rigidity material in slip-strengthening conditions, so there may not be any aftershocks on the shallow fault. Focal mechanisms of larger shallow aftershocks (Supplementary Fig. 20) show a variety of oblique strike-slip and extensional mechanisms

with distinct orientations that indicate complex shallow stress, but do not indicate trench-parallel compression. The fast-slip faulting induces moderate positive Coulomb stress changes on the slow-slip fault orientation (Supplementary Fig. 20), compatible with triggering of the slow-slip. On-land geodetic and InSAR data are unable to resolve the proposed faulting as we have shown for the nearby hr-GNSS stations. We do not detect any obvious feature from available bathymetry maps that could corroborate the slow-slip faulting geometry, although diffuse northwest trending structures in the Beringian margin disrupt the western ends of the accreted terranes a few tens of kilometers to the west[28]. The nearby seismic profiles that are available are trench-perpendicular transects, so they are not sensitive to the proposed faulting geometry. Analysis of campaign seafloor geodetic observations from before and after the event may enable further constraints to be placed on the process, but the contribution from the 22 July 2020 mainshock and any deep afterslip, and the fast-slip component of the 19 October 2020 aftershock must be accounted for.

Figure 10 shows the regions that have been inferred to have strong geodetic coupling and weak geodetic coupling, which may play an important role in the lateral shearing within the Pacific plate[15], but there is very little resolution of the shallow megathrust coupling along the 1938 and 2021 Semidi ruptures or along the Shumagin segment. Seafloor geodesy may help to resolve whether there is strain release or a lateral gradient in strain accumulation on the megathrust near the 19 October 2020 event. This information is needed to understand the cause of lateral compression in the upper wedge implied by our slow slip source. If the process instead involved slumping across the shelf break rather than slow thrusting within the wedge, high-resolution bathymetric scans may help to resolve the occurrence of such mass wasting, but as we discuss, it is challenging to have substantial slumping go undetected by the nearby geodetic stations. Dense reflection profiling might resolve the faults involved in this complex event, and complex structures have been indicated in existing sparse profiles[17], but 3D imaging is likely needed to resolve structures with a strike close to perpendicular to the ridge.

Tsunamigenic slow-slip rupture on non-splay faults in the upper plate should be considered as an additional potential tsunami hazard, adding to that for slow-slip on the shallow megathrust or on splay faults that have been associated with tsunami earthquakes. For the 19 October 2020 event, upper wedge deformation provides a viable explanation for how the event generated much larger amplitude tsunami signals than were produced by a larger thrust faulting event deeper on the megathrust. Whether any slumping contributed is yet to be determined, but geometrically does not seem favorable. Such upper wedge deformation may involve complex structures from the accretionary history of the wedge and motivates high resolution 3D imaging of shallow prisms to detect potential shallow faulting geometries.

## Methods
### Data processing
We select 62 *P* and 50 *SH* broadband recordings from the Incorporated Research Institutions for Seismology (IRIS) data management center with well-distributed azimuthal coverage at teleseismic epicentral distances between 30° and 90° (station distributions and data are shown in Supplementary Fig. 3). Instrument responses are removed to obtain ground velocities in the passband 1–300 s with waveform durations of 100 s. We precisely aligned *P* and *SH* wave initial motions manually.

We selected waveforms from 9 regional broadband stations and 6 local strong-motion stations at epicentral distances <700 km (Supplementary Fig. 4). The recordings are converted to ground velocities by removing the instrument responses, and all waveforms are filtered with a period band of 5–100 s. The ground velocity records are re-

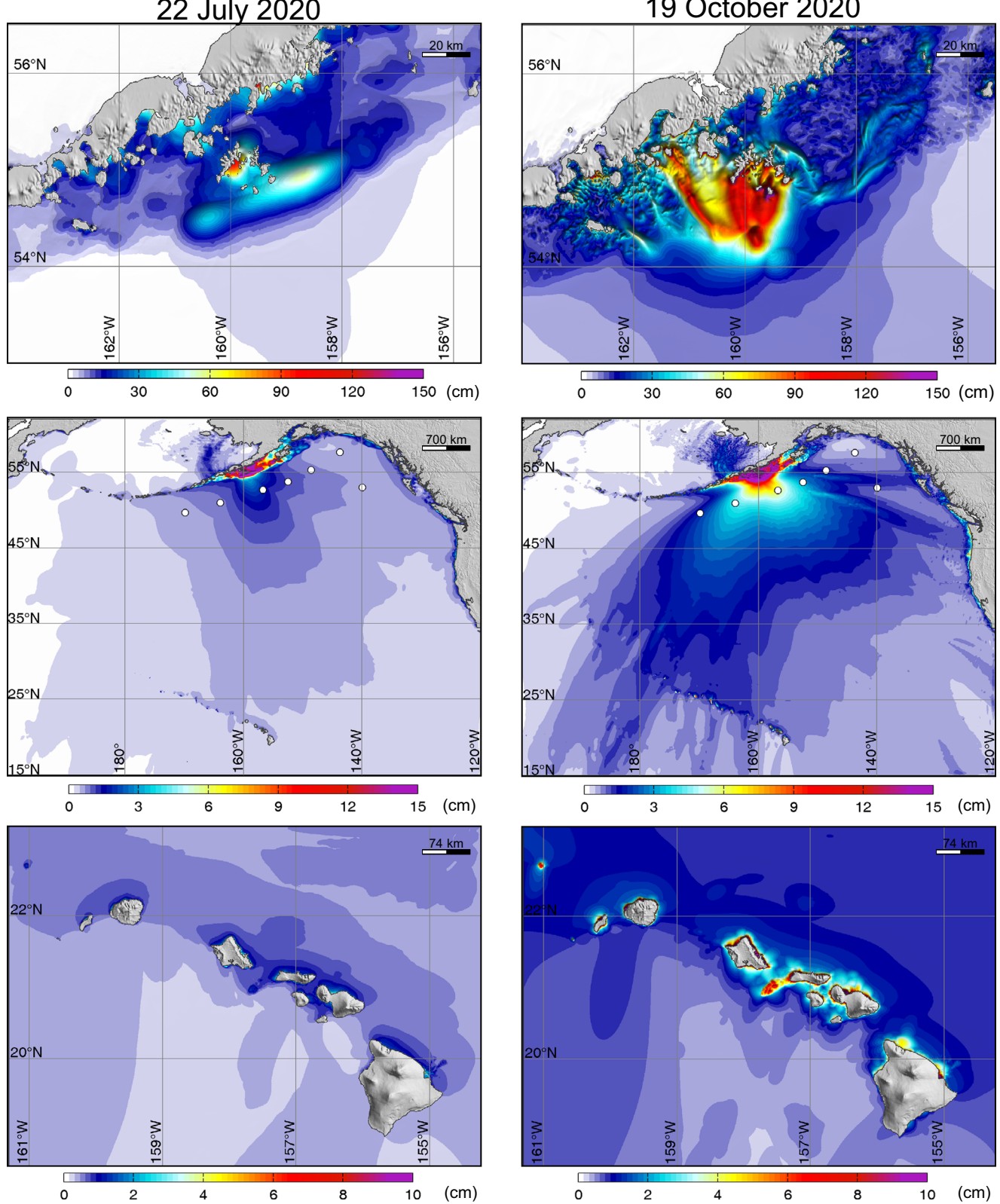

**Fig. 9 | Comparison of the computed tsunami maximum sea elevation for the 22 July 2020 and 19 October 2020 events.** Computed near-field (top row) and far-field (middle and lower rows) tsunami wave amplitude for the 22 July 2020 $M_W$ 7.8 mainshock using a finite-source model[6] (left column) and for the preferred three-fault model for the 19 October 2020 $M_W$ 7.6 aftershock (Fig. 7) (right column). White circles denote locations of the DART stations.

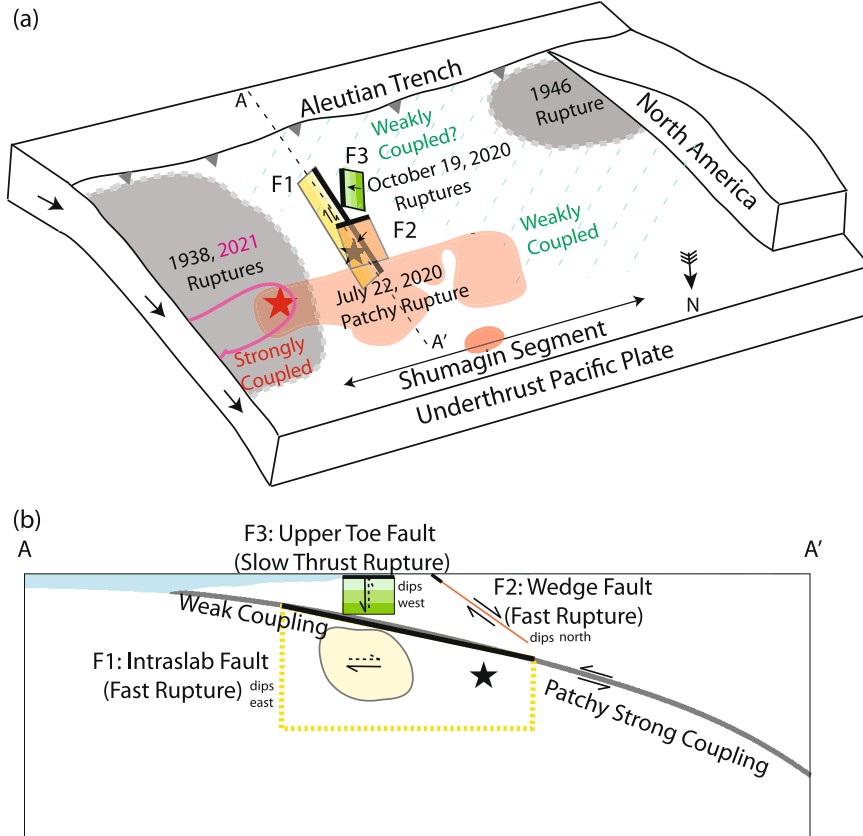

**Fig. 10 | Model summary. a** Schematic map and (**b**) vertical cross-section of the Shumagin Segment region indicating geometry of the preferred model with three faults that ruptured in the 19 October 2020 $M_W$ 7.6 event (black star indicates the hypocenter): F1 has fast strike-slip rupture within the underthrust Pacific plate (yellow zone); F2 has fast oblique-normal faulting rupture in the overthrusting North America plate (orange zone); F3 has slow thrust faulting rupture on a fault in the upper plate (green zone). The shallow edges of the fault models are indicated by thick lines. The sequence ruptured seaward of the 22 July 2020 $M_W$ 7.8 interplate thrust event (pink region with ≥0.5 m slip[6]; red star indicates the hypocenter). Megathrust coupling variations are indicated, along with aftershock zones of the 1938 and 1946 megathrust events (gray patches), and ≥0.5 m slip zone of the 2021 Chignik megathrust event[16] (magenta outline).

sampled with a time interval of 0.2 s. We hand-picked the *P* wave first arrivals, and a time window of 180 s was used for the joint inversion. Eight coseismic static displacements from GNSS sites processed by the Geodesy Laboratory at Central Washington University were used (Fig. 1). Eight hr-GNSS time series which include both seismic arrivals and static offsets were also used, with data provided by the UNAVCO website (Supplementary Fig. 5).

### Finite fault inversion
As seismic and geodetic data can provide complementary constraints on the rupture process, we used both data types to invert the rupture process of the 19 October 2020 event assuming first one and then two fault segments. We performed non-linear finite fault inversions[29,30], involving the joint analysis of coseismic static offsets, hr-GNSS time series, and seismic waveforms. A simulated annealing algorithm was used to solve for the slip magnitude and direction, rise time, and average rupture velocity for subfaults on the two segments. For each parameter, we set specific search bounds and intervals. The subfault size is chosen as 5 km × 5 km, and the rake angles on the two fault segments are constrained to be right-lateral purely strike-slip and purely dip-slip, respectively. We allowed both the rise and fall intervals of the asymmetric slip rate function for each subfault to vary from 0.6 to 6.0 s; thus, the corresponding slip duration for each subfault is limited between 1.2 and 12 s. We let the slip vary from 0.0 to 8.0 m, and the average rupture velocity is allowed to vary from 0.5 to 3.0 km/s. Green's functions for static displacements and seismic waveforms are computed using a 1-D layered velocity model[31]. Equal weighting among the data functionals for GNSS statics and seismic waveforms was used in this study.

### Tsunami modeling
NEOWAVE is a non-hydrostatic model utilizing a vertical velocity term to account for dispersion properties comparable to low-order Bous-sinesq-type equations[32]. The vertical velocity term also facilitates modeling of flows on steep continental slopes and tsunami generation from seafloor deformation over a finite rise time. These dynamic processes are important for resolution of developing tsunami waves in the near field and accurate reproduction of the DART records. The time history of seafloor vertical displacement at the source is determined from the finite-fault model using an elastic half-space solution[33] and augmented by the horizontal motion of the seafloor slope[34]. The numerical solution is obtained by the finite difference method with nested computational grids in spherical coordinates. The nesting scheme includes two-way communications during the computation and does not require an external transfer of data between grid layers.

Four levels of telescopic grids are needed to model the tsunami from the sources with increasing resolution to the Kahului tide gauge. An additional level is needed to resolve the more complex waterways leading to Hilo, King Cove, and Sand Point. Supplementary Fig. 7 shows the layout of the computational grid systems. The level-1 grid extends across the North Pacific at 2-arcmin (~3700 m) resolution, which gives an adequate description of large-scale bathymetric features and optimal dispersion properties for modeling of trans-oceanic tsunami propagation with NEOWAVE[35]. The level-2 grids resolve the insular

shelves along the Hawaiian Islands at 24-arcsec (~740 m) and the continental shelf of the Alaska Peninsula at 30-arcsec (~925 m), while providing a transition to the level-3 grids for the respective islands or coastal regions at 6-arcsec (~185 m) resolution. The finest grids at levels 4 or 5 resolve the harbors where the tide gauges are located at 0.3-arcsec (9.25 m) or 0.4 arcsec (12.3 m). A Manning number of 0.025 accounts for the sub-grid roughness at the harbors. The digital elevation model includes GEBCO at 30-arcsec (~3700 m) resolution for the North Pacific, multibeam and LiDAR data at 50 m and ~3 m in the Hawaii region, and NCEI King Cove 8/15-arcsec dataset and Sand Point V2 1/3-arcsec dataset, which also covers the Shumagin Islands.

### Long-period spectral analysis

Global recordings of broadband ground motion from stations of the Global Seismic Network and Federation of Digital Seismic Networks were collected for the 19 October 2020 earthquake. Ground displacements of long-period fundamental mode Rayleigh Waves and Love (G) Waves were group-velocity windowed for short-arc (R1, G1) and long-arc (R2, G2) arrivals and their spectra were computed. The spectral measurements at a period of 256 s were corrected for propagation back to the source epicenter using phase velocity and attenuation values from the Preliminary Reference Earth Model (PREM)[36], and the amplitude spectra are plotted versus azimuth from the source in Supplementary Fig. 6. Calculations for point-source representations of each of the 3-fault model components were then made using PREM excitation functions, and the individual and 2-fault sums are plotted on the data in Supplementary Fig. 6.

For the intraslab fast-slip strike-slip fault, computations use seismic moment $M_0 = 2.43 \times 10^{20}$ Nm, strike 350°, dip 50°, rake 173°, and depth 35.5 km. For the upper plate fast-slip oblique normal fault, computations use $M_0 = 0.29 \times 10^{20}$ Nm, strike 260°, dip 35°, rake 225°, and depth 15 km. For the upper plate slow-slip thrust fault, computations use $M_0 = 1.8 \times 10^{20}$ Nm, W = 20 km, L = 20 km, slip 15 m, strike 190°, dip 30°, rake 90°, and depth 8 km. The rigidity used for the strike-slip faulting was 5.4 GPa, and it was 3.2 GPa for the oblique faulting and 3.0 GPa for the thrust faulting.

### Slow megathrust rupture

A plate boundary thrust-fault model for the additional source of tsunamis involves a compact 20 km × 20 km slip patch with an upper edge 22 km deep, and strike 250°, dip 12°, and rake 90°, with 16 m of pure thrust slip. The slow-fault ruptures 30 s after the initiation of the earthquake and lasts for 5 min. The time-varying seafloor deformation of the slow-slip event is approximated by the Okada solution at each computational time step together with those from the fast-slip event, and the associated evolution of the tsunami is dynamically and internally resolved by NEOWAVE driven by the prescribed kinematic seafloor conditions to fit the DART records. Assuming a rigidity of 30 GPa, appropriate for the shallow megathrust environment, the seismic moment is $1.92 \times 10^{20}$ Nm ($M_W$ 7.46). The computed seafloor deformations for the two-fault coseismic rupture and the delayed slow slip on the thrust patch are shown in Supplementary Fig. 8, separately and combined. The thrust slip patch is located near the shelf break and similar to the dipole fitting has about 20 km absolute uncertainty, but cannot locate significantly out onto the continental slope, as the tsunami excitation changes rapidly along the slope and incompatible waveforms are produced at the DART stations. The resulting seafloor deformation resembles a scaled-up version of the 2-fault model with uplift and subsidence straddled across the shelf break. Comparisons of the observed and computed tsunami signals for the three-fault model are shown in Supplementary Fig. 9, with clear evidence of uniform improvement relative to the two-fault solution in Fig. 4. The fits are slightly improved in comparison to those for the optimal dipole model in Fig. 6. The large second arrival and the following trough in the DART waveforms are matched well by the slow-slip event. The tide gauge

records, which were not used in the source deduction, provide independent validation of the model results. In particular, the computed tsunami waves from the two sources are out-of-phase in Hawaii waters and the matching with the tide gauge records through destructive interference is remarkable (Supplementary Fig. 9). We reject this specific model despite its ability to match the tsunami data because it predicts larger dynamic displacements at GNSS stations AC12 and AC28 (Supplementary Fig. 8), which are not observed after the motions from the fast rupture.

### Slow splay fault rupture

An upper plate splay-fault model for the additional source of tsunami waves involves a compact 20 km × 30 km slip patch with an upper edge 3 km deep, and strike 250°, dip 35°, and rake 90°, with 12 m of pure thrust slip. The slow-fault ruptures at the same time as the initiation of the earthquake and lasts for 5 min. Assuming a rigidity of 30 GPa, appropriate for the shallow megathrust environment, the seismic moment is $2.16 \times 10^{20}$ Nm ($M_W$ 7.49). The computed seafloor deformations for the two-fault coseismic rupture and the slow thrust slip on the splay patch are shown in Supplementary Fig. 10, separately and combined. The thrust splay patch is located near the shelf break and similar to the dipole fitting has about 20 km absolute uncertainty, but cannot locate significantly out onto the continental slope, as the tsunami excitation changes rapidly along the slope and incompatible waveforms are produced at the DART stations. The resulting seafloor deformation again resembles a scaled-up version of the 2-fault model with uplift and subsidence straddled across the shelf break. Comparisons of the observed and computed tsunami signals for the three-fault model are shown in Supplementary Fig. 11, with clear uniform improvement relative to the two-fault solution in Fig. 4. The fits are slightly improved in comparison to those for the optimal megathrust slow-slip model in Supplementary Fig. 9. The large second arrival and the following trough in the DART waveforms are matched well by the slow-slip event. The computed tsunami waves from the two sources are out-of-phase in Hawaii waters and the matching with the tide gauge records through destructive interference is remarkable (Supplementary Fig. 11). Again, we reject this specific model despite its ability to match the tsunami data because it predicts larger dynamic displacements at GNSS stations AC12 and AC28 (Supplementary Fig. 10), which are not observed after the motions from the fast rupture.

### Coulomb failure stress

The coulomb failure stress changes can be written as $\Delta CFS = \Delta\tau + \mu'\Delta\sigma N$[37], where $\Delta\tau$ and $\Delta\sigma N$ denote the shear stress and normal stress change on the receiver fault. The parameter $\mu'$ is the effective coefficient of friction on the fault and is set as 0.4 in this study. Using the slow-slip faulting as the receiver fault (Strike 190°, dip 30°, rake 90°), we computed the $\Delta CFS$ at different depths caused by the fast rupture of two intraplate faults (Supplementary Fig. 20).

## Data availability

Coseismic GNSS displacements and hr-GNSS time series were obtained from the UNAVCO Bulletin Board (https://www.unavco.org/data/gps-gnss/gps-gnss.html). The GNSS data are based on services provided by the GAGE Facility, operated by UNAVCO, Inc., with support from the National Science Foundation and the National Aeronautics and Space Administration under NSF Cooperative Agreement EAR-1724794. We also thank the CSRS-PPP online service system for hr-GNSS data processing (https://webapp.geod.nrcan.gc.ca/geod/tools-outils/ppp.php). Teleseismic body wave and regional broadband records were obtained from the Federation of Digital Seismic Networks (FDSN: https://doi.org/10.7914/SN/IU, https://doi.org/10.7914/SN/II, https://doi.org/10.7914/SN/CN, https://doi.org/10.18715/GEOSCOPE.G, https://doi.org/10.7914/SN/CU, https://doi.org/10.7914/SN/IC, https://doi.org/10.7914/SN/AV, https://doi.org/10.7914/SN/AK, https://doi.

org/10.7914/SN/TA), and accessed through the Incorporated Research Institutions for Seismology (IRIS) data management center (http://ds.iris.edu/wilber3/find_event). We thank the facilities of IRIS Data Services, and specifically the IRIS Data Management Center, which were used for access to waveforms, related metadata, and/or derived products used in this study. IRIS Data Services are funded through the Seismological Facilities for the Advancement of Geoscience (SAGE) Award of the National Science Foundation under Cooperative Support Agreement EAR-1851048. Strong-motion recordings were obtained from the Center for Engineering Strong Motion Data (CESMD, https://strongmotioncenter.org/). Earthquake information is based on the catalogs from the U.S. Geological Survey National Earthquake Information Center (USGS-NEIC) (https://earthquake.usgs.gov/earthquakes) and the Alaska Earthquake Center (http://earthquake.alaska.edu), last accessed July 13, 2022. The high-resolution digital elevation model, Sand Point V2, at the Shumagin Islands was downloaded from the National Centers for Environmental Information (https://maps.ngdc.noaa.gov/viewers/bathymetry/).

## Code availability

Codes for kinematic slip inversion and for tsunami modeling may be requested from the authors.

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

## Acknowledgements

Y.B. was supported by the Finance Science and Technology Project of Hainan Province (No. ZDKJ202019). C.L. was supported by the National Science Foundation of China (No. 42222403). T.L. was supported by the National Science Foundation under grant number EAR-1802364.

## Author contributions

Y.B. performed all tsunami modeling calculations. C.L. performed all finite fault inversions. T.L. conceived the study and contributed to model developments. K.C. contributed to model developments and tsunami generation insights. Y.Y. contributed to grid specification and initial model set-up. All authors contributed to writing the paper and interpretation of the results.

## Competing interests

The authors declare no competing interests.
