## [Peer Review File · Nature Communications]

REVIEWER COMMENTS

Reviewer #1 (Remarks to the Author):

Bai et al. investigate the source process of the October 2020 Shumagin Islands intraslab earthquake based on seismic, geodetic, and tsunami records. They find that a second, minor fault rupture is required to explain the seismic and geodetic records, but the two combined "fast-faulting" sources still cannot fully explain the large tsunami. They search for a slower source that could produce the observed tsunami, concluding that it occurred updip of the main earthquake, near the ocean floor. They propose a thrust faulting mechanism on a fault perpendicular to the trench, which explains the data but is highly non-unique. This study is very interesting for how it systematically addresses what is needed to account for various observations, and I think it is important to see it published eventually for how the tsunami reveals something unexpected and otherwise unobserved. However, I would like to see its results presented in a more honest, robust way. Right now, there is a level of confidence in how the authors present their fault source model that I think obscures the uncertainty inherent in their assumptions. I think that these need to be addressed comprehensively before publication.

Major Comments

Throughout the manuscript, the authors have a tendency to be too qualitative when discussing their results. I provide some examples below, but it is not a comprehensive list because I did not want the review to be too long. In some cases, I find the qualitative statements to be effective for the narrative of the document, especially when they are immediately followed up with a more specific, quantitative statement. However, this only occurs a couple of times (understandably so in a short-format paper). In other cases, the authors lean on figures to make a point, but I cannot always easily see what is being specified from the figures. I think the solution to all of these issues is to make up-front statements – particularly in the Results and Discussion section – that are generally more precise and quantitative.

Although I fully support the authors' conclusion that a second fault (or at least a change in fault geometry) is needed to explain the seismic and geodetic observations, the justification for the location and orientation of the second fast-faulting component (p. 4, ll. 11ff.) seems weak, if not unconvincing. First of all, contrary the authors' statement on p. 5, ll. 9-10, I would argue that most of the aftershocks, and the vast majority of aftershock seismic moment, in the vicinity of the October 2020 event can be interpreted to be within the slab, certainly considering depth uncertainties. It would be surprising to me to see such a significant ($M_w \sim 7$) component of the earthquake highlighted by so few aftershocks. Could this normal faulting have occurred within the slab and still explain the seismic and geodetic observations? Secondly, the authors are guided by a major landward-dipping normal fault imaged in the forearc, but then proceed to choose the auxiliary plane to this fault as their preferred geometry. It seems unlikely that such a major fault (3-4 meters of slip in this event) would (a) be invisible and (b) work mechanically if it intersected the fault seen in the geophysics. I would prefer a landward-dipping fault for this reason, despite the better fits (that the authors do not quantify) produced by the seaward-dipping fault.

Following up on the previous point, although each step of the authors' analysis is based on datasets and observations, there are assumptions inherent in each step as well. I would argue that the authors' results, particularly those for the final, slow faulting source, are more plausible than certain, and the authors should emphasize this point substantially more.

Minor Comments

The state of shallow megathrust coupling is an important issue in subduction zone earthquake science, which the authors explicitly recognize in the Introduction (pp. 2-3). However, as far as I can tell, they do not return to the issue of megathrust coupling in their

Discussion. If their analysis cannot reveal anything about the shallow coupling on the megathrust, that is fine, but then I would prefer they talk less about it in the Introduction.

One of the most important sections is about the poor fit to tsunami using the fast-slip model that fits the seismic and geodetic observations (p. 5, ll. 15ff.). I understand that Nature Communications is a short format journal, but in my opinion this section is too important to be given such little space. I would expand it with more detail if possible.

Detailed Comments

p. 1, l. 5: The phrasing in the first sentence of the Abstract is unclear and sets a vague framework for the context of this study. Were slow, shallow ruptures not considered hazardous prior to this paper? I might just remove this sentence and start with the next sentence.

p. 1, ll. 13-14: I think it is important to indicate the uncertainty of the slow, tsunamigenic source somewhere in the Abstract, indicating that it is a plausible scenario consistent with the datasets investigated here.

p. 2, l. 14: It might be useful to provide a sense of hypocenter depth uncertainty. I suspect that most of these events could have occurred within the Pacific slab.

p. 2, ll. 14-16: I would add that most large, intraslab ruptures are under or seaward of the trench, not 50+ km arcward of the trench

p. 4, l. 8: How much worse is the fit when slip is confined to larger depths?

p. 4, ll. 11ff: In this paragraph, I think that statements like "fitting the teleseismic data better," "concentrates slip within the fault model," "a more stable and sensible solution," and "typically fast" are too qualitative

p. 5, ll. 12-13: Although potentially true, this last sentence is too speculative to be useful. I would remove it or run a quick back-slip model with a transition in coupling (easy enough to do with a similar setup as the Coulomb stress modeling) and resolve the stresses on the second fast-slip component to see if slip on that structure is favored. That way, the authors could be justified in making the assertion that this specific faulting is favored by a transition in coupling.

p. 6, ll. 17-20: One thing that might help me in this discussion is to label one or more of the tsunami waveforms in Fig. 4 with the impulsive peak and the wide trough. That might help make this result more concrete. In general, I think that the authors could leverage their nice figures even more in the text.

p. 7, l. 8: How large is the "large time discrepancy?"

p. 7, l. 20: "to account for run-out effects while conserving volume" should have a reference and/or explanation

p. 8, ll. 3-6: These first two sentences of this paragraph seem like they should come at the beginning of the section.

p. 8, ll. 7ff: There are some vague or qualitative statements in here that could be made more precisely. What is the azimuth covered by the DART buoys? Is dipole orientation also a factor in tsunami waveform phase, or mostly location? What does it mean for seafloor motion to be "sensitively determined by finding consistent and accurate arrival time at each DART?" What were the parameter ranges in the search? What tradeoffs are there in the parameters?

p. 9, l. 12: I think the authors could more explicitly set up why they prefer faulting to a landslide/slump source

pp. 9-10: I think these sections could be merged without a significant loss of clarity

p. 11, l. 1: It seems that a statement is needed to introduce the subsequent discussion such as, "Assuming this unusual, unobserved, slow faulting is the source of the tsunami..."

p. 11, ll. 3-4: This is another speculative statement. Remove these or support them with evidence.

p. 13, l. 14 and l. 19: Maybe it is convention for the data types, but it seems strange to me to report frequency and period bands right next to each other

p. 14, ll. 5-6: Are the faults constrained to have no obliquity at all? Also, "right lateral strike slip" and "normal slip" are not strictly speaking "rake angles."

p. 14, ll. 11-12: Equal weighting can take some different forms, so I would prefer to see an equation for calculating misfit

p. 16, ll. 22ff: I am not fond of the repetition in this section from the previous section. At first I thought the authors had made a mistake in their writing.

Fig. 1: I think the cross-section should have the same orientation as in Fig. 10.

Fig. 3: The image should use the full range of the color scale, unless using exactly the same color scale in another figure (which is not the case, because the scale in Fig. 7 has a different range). Also, please modify the figures to use color-blind friendly color maps (not rainbow, jet, lots of reds/greens, etc.)

Fig. 7: This is a very complicated paneled figure, and might need to be broken into parts. It is so dense that I thought some of the panel labels referred to other adjacent panels. I think the GNSS timeseries should zoom into the 0-100 second range and ignore the (mostly zero) time series from 100 to 600 seconds.

Fig. 9: This does not seem to be necessary as a primary figure, although I do like it (with a better color map).

Reviewer #2 (Remarks to the Author):

Bai and colleagues present a combined analysis of GNSS, seismic and tsunami data to understand an enigmatic M7.6 earthquake that occurred in the Alaska subduction zone in October 2020. Unlike other recent, large earthquakes in this region, this event was an intraplate event. The authors convincingly show that slip on structures in the overriding and subducting plates is required to explain all of the observations, including a secondary tsunami source. This paper is significant because it offers the opportunity to investigate complex interactions between faults in the subducting and overriding plates and their implications for seismic and tsunami hazards. Overall, I found the paper well written and illustrated. I am not an expert in the types of analyses presented, but I generally found the analyses clearly described and convincing. I have a few comments and questions that I think can be mostly addressed with minor revisions to the manuscript.

The authors require slip on two faults in the overriding plate to explain their observations. One of my primary comments is that I think the paper would be much stronger if the authors considered other geological/geophysical evidence for faulting in the overriding plate to support and contextualize their observations. I support the author's approach of first guiding their definition of ideal fault geometries and slip based on the analysis of the seismic, geodetic and tsunami data, alone. However, given the understandable uncertainty in solutions, the manuscript would be stronger if then provided geological and geophysical data to support their results and potentially guide the choice of preferred solutions. I provide specific comments/suggestions on this and other points below.

-The authors consider two geometries for a fault in the upper plate that they require to explain the non-double couple component of the M7.6 earthquake: one is south dipping and the other north dipping. The authors prefer the south dipping fault because of a better fit to the data (pg 4, Line 19). However, a north dipping structure might be more geologically plausible based on the fact that the upper plate of this subduction zone comprises a series of accreted terranes separated by north dipping sutures (Horowitz et al., 1989; Rowe et al., 2011) and that a major north dipping fault has been imaged in seismic reflection profiles (Bécel et al., 2017, von Huene et al., 2019), which may have reactivated one of these collisional sutures (Shillington et al., 2022). The difference in data fit between the north and south dipping upper plate fault is not quantified in the paper. How much better is the fit for the south dipping fault than the north dipping fault, and is it significant in light of uncertainties? Given that a south dipping fault is less consistent with other existing observations, I think the paper needs to make a stronger case for preferring this geometry. If the difference in misfit is not significant, I think that a north-dipping fault is more consistent with what we know about the upper plate of this margin.

-If a north dipping fault is used to explain the fast rupture component of this event, what are the implications for the location and geometry of the secondary tsunami source?

-The section entitled "Constraining the secondary tsunami source" could be written more clearly and succinctly. I found some of the text repetitive (e.g., pg 8, Line 3-8). On the other hand, some of the logic in this section was not clear to me as written. If I understand, the dipole modeling approach used to narrow down possible locations for the secondary source could apply to either a shallowly dipping fault or to a submarine land slide. However, the authors appear to rule out a landslide as a possible secondary tsunami source and only consider slow-faulting scenarios in the rest of the paper. This choice and supporting logic should be laid out more explicitly (e.g., by rewriting and expanding Lines 7-11, pg 9, if that is the rationale).

-The interpretation of a trench-normal upper plate thrust fault experiencing slow slip is very interesting, and its quite remarkable if such a fault would produce no seismic or geodetic signals. The paper would be stronger if it explored other evidence for this fault (e.g., Lines 5-10, Pg 12). Is there any other possible dataset that could have detected such a slow slip event (e.g., InSAR given the proximity to the Shumagin Islands)? Is there any other evidence of trench normal compression (e.g., from earthquake focal mechanisms, etc)? Or other geological support for such a fault? This region is near (although east of) the edge of the Beringian margin, where trench oblique structures are observed – could the orientation of this structure plausibly parallel those? I think the paper would be stronger if there was more discussion about this feature since it is a major conclusion of the paper and a very surprising one. It is not very satisfying that the paper proposes this extraordinary fault, but does not discuss it in much detail.

Other minor comments:

-Pg 7, Lines 13-15: I am not familiar with this type of parameterization, so the description of a trough, ridge and level arm was not clear to me. Suggest rephrasing or elaborating a bit more (or adding a figure to the supplement to illustrate).

-Figure 10 labels the updip portion of the megathrust as weakly coupled even though the text correctly states that coupling on the offshore, shallow megathrust is poorly known (pg 2, Lines 23-25). Is a downdip change in coupling important to the results presented here?

Thank you for the opportunity to review this interesting paper,
Donna Shillington

Reviewer #3 (Remarks to the Author):

(Jeff Freymueller)

This paper addresses a lingering mystery of the October 2020 Sand Point earthquake: why this strike-slip (or mainly strike-slip) earthquake that occurred primarily in the downgoing oceanic plate generated a larger tsunami than the larger, megathrust July 2020 Simeonof earthquake that preceded it.

The authors first develop a model, which they refer to as the "fast slip" model, based on seismic waveforms and GNSS displacements. Their model is more complex than previously published models, and includes fault slip both above and below the plate interface. This model is well described, and fits the data used to derive it but does not explain the resultant tsunami. (Nor has anyone else's model fit the tsunami). This leads them to consider a secondary tsunami source, which needs to start 4-5 minutes after the mainshock to explain the DART buoy record. The challenge here is that one key constraint on this secondary seismic source is that it must not produce any seismic or geodetic signature. As a result, most straightforward fault models don't work because they would produce non-zero displacements at the GPS sites.

The authors finally propose that the secondary tsunami source was slow rupture on a thrust fault on the continental shelf that strikes nearly trench normal. The magnitude of slip is very high (17 meters), and the resulting seismic moment is nearly as large as the "fast slip"

earthquake model itself. The source is oriented so that the main seafloor displacement occurs in the right place, and that there are no displacements observed at the GPS sites. Lower slip on a larger fault might fit the tsunami data, but it would be harder for such a fault to remain invisible to the geodetic data and the authors note that it might not explain the tsunami data as well. The rupture was presumed to be slow enough to be invisible to seismometers. Is it really true that a ~ 300 sec slow rupture of $M \sim 7.5$ would leave no seismic signature at all? Might there not be some long-period signal associated with that? Substantially smaller events resulting from glacial sources have in fact been detected. However, I could certainly be wrong about the source duration of those events.

As far as I can tell, the model does fit the data constraints. I still find it hard to believe, though. It is intrinsically hard to dispel all skepticism about a component of a model source when one of the primary constraints is that it must be invisible to the data sets used to develop the slip distribution. I also have to wonder why such a fault would exist in the first place – is there any evidence for it? – and what would make it able to slip such an extraordinary amount on a relatively small fault patch? What would load a fault of this orientation to bring it close to failure? As a result, this model seems to me to be highly speculative.

What I would like to see from the authors is more support for how this is the only possible explanation of the tsunami. Unless that is demonstrated, then all they can really do is put this forward as a possible explanation that doesn't violate any data we now have (there is also a GPS-Acoustic seafloor displacement that will become available, but it is not yet published). Here are my key remaining questions:

Would such a large event with ~ 300 sec duration actually be invisible to long-period seismology? Can you back up such an assertion more thoroughly?

Is there any evidence in past seismicity, bathymetry, or mapped faults for the structure that they have proposed?

Why would a structure be tectonically loaded, given that it is nearly orthogonal to plate convergence?

Large slip has been observed in tsunami earthquakes on the very shallow megathrust, but this proposed source is not on the megathrust; it would cut what we presume to be normal forearc upper crust. Can they point to any other examples of very large slow slip on a fault in a non-megathrust setting?

Overall, the writing and organization of the paper is very good, so there are almost no minor corrections. Just one:

Page 9, line 24. Note that a long source process may obscure the seismic expression, but it will not obscure the (static) geodetic expression of the deformation due to the slip. Also, as noted above, I would like to see a stronger argument that such a large moment event would remain invisible at long periods (it may be that it would, but the assertion should be backed up given that long-period sources of magnitude much smaller than this have been detected and reported).

Response to reviews of "Fast and slow intraplate ruptures during the 19 October 2020 magnitude 7.6 Shumagin earthquake" by Bai et al. The reviewer comments are reproduced below in black type, and our responses and indications of how we have revised the manuscript to address the comments are shown in blue.

REVIEWER COMMENTS

Reviewer #1 (Remarks to the Author):

Bai et al. investigate the source process of the October 2020 Shumagin Islands intraslab earthquake based on seismic, geodetic, and tsunami records. They find that a second, minor fault rupture is required to explain the seismic and geodetic records, but the two combined "fast-faulting" sources still cannot fully explain the large tsunami. They search for a slower source that could produce the observed tsunami, concluding that it occurred updip of the main earthquake, near the ocean floor. They propose a thrust faulting mechanism on a fault perpendicular to the trench, which explains the data but is highly non-unique. This study is very interesting for how it systematically addresses what is needed to account for various observations, and I think it is important to see it published eventually for how the tsunami reveals something unexpected and otherwise unobserved. However, I would like to see its results presented in a more honest, robust way. Right now, there is a level of confidence in how the authors present their fault source model that I think obscures the uncertainty inherent in their assumptions. I think that these need to be addressed comprehensively before publication.

We appreciate the reviewer's interest in the challenging analysis. We address the model uncertainty issues to a greater level in the revision, both for the secondary fast-slip faulting and the slow-slip faulting, to better convey the limitations and intrinsic demands of finding a source model for a strong tsunami with attributes that are undetected by conventional seismic and geodetic means. We now include results for a range of models, each being optimized to fit the data as well as possible, to better convey the basis for our preferred, albeit non-unique model.

Major Comments

Throughout the manuscript, the authors have a tendency to be too qualitative when discussing their results. I provide some examples below, but it is not a comprehensive list because I did not want the review to be too long. In some cases, I find the qualitative statements to be effective for the narrative of the document, especially when they are immediately followed up with a more specific, quantitative statement. However, this only occurs a couple of times (understandably so in a short-format paper). In other cases, the authors lean on figures to make a point, but I cannot always easily see what is being specified from the figures. I think the solution to all of these issues is to make up-front statements – particularly in the Results and Discussion section – that are generally more precise and quantitative.

We adopt this recommended approach to the extent possible and clarify the relevant information in the figures supporting each of our findings. We do this during discussion of the modeling as well as in the Results and Discussion section, as suggested. We augment the figure labeling to help the reader connect with the text.

Although I fully support the authors' conclusion that a second fault (or at least a change in fault geometry) is needed to explain the seismic and geodetic observations, the justification for the location and orientation of the second fast-faulting component (p. 4, ll. 11ff.) seems weak, if not unconvincing. First of all, contrary the authors' statement on p. 5, ll. 9-10, I would argue that most of the aftershocks, and the vast majority of aftershock seismic moment, in the vicinity of the October 2020 event can be interpreted to be within the slab, certainly considering depth uncertainties. It would be surprising to me to see such a significant ($M_w \sim 7$) component of the earthquake highlighted by so few aftershocks. Could this normal faulting have occurred within the slab and still explain the seismic and geodetic observations? Secondly, the authors are guided by a major landward-dipping normal fault imaged in the forearc, but then proceed to choose the auxiliary plane to this fault as their preferred geometry. It seems unlikely that such a major fault (3-4 meters of slip in this event) would (a) be invisible and (b) work mechanically if it intersected the fault seen in the geophysics. I would prefer a landward-dipping fault for this reason, despite the better fits (that the authors do not quantify) produced by the seaward-dipping fault.

We re-examined the second fast-faulting component in greater detail, considering a wide range of models with either south-dipping or north-dipping faults in the upper plate or in the slab. We find that the resolution is predictably limited, as this is a minor (10%) component of the total fast-faulting radiation and all models require that it occur in the later part of the moment rate function, so depth resolution is limited. With our more extensive search over the secondary faulting location and depth, we find that the north-dipping seismically-imaged normal fault in the upper plate is at least an equally viable model for fitting the data and we adopt that model as the preferred case. The reason for this preference over an intraslab north-dipping fault or a south-dipping upper plate fault (the solutions for the updated optimal versions of both of these cases are shown in the Supplement, and they provide comparable fits to the data) is because the north-dipping normal fault is independently known to exist, and mechanically it is more sensible, as noted by the reviewer. In terms of fitting the seismic and geodetic data, deeper faults within the slab can fit the data comparably, but the geometry would require rupturing across the primary strike-slip fault and the slip distribution is bimodal, at the ends of the secondary fault, which suggests a poorly resolved slip. We make it clear that our preferred model is not uniquely resolved, and represents a decision made on both reasonable slip distribution and consistency with a known fault structure. We modify summary description of the final model to provide clearer basis for our preferred model, along with demonstrating that our choice of the secondary fast-rupturing fault geometry does not at all impact our modeling of the slow-slip component.

Following up on the previous point, although each step of the authors' analysis is based on datasets and observations, there are assumptions inherent in each step as well. I would argue that the authors' results, particularly those for the final, slow faulting source, are more plausible than certain, and the authors should emphasize this point substantially more.

We modify our description of the decisions involved in selecting a preferred model to more accurately convey the non-uniqueness and alternate possibilities allowed by the data constraints. While the waveform modeling is non-unique, as we clearly state in the revision, the tsunami waveform fitting does provide some fundamental constraints on the seafloor motion that produced it, irrespective of detailed orientation of the faulting. The revision makes it clear that our solution demonstrates existence of a viable solution, but is not a unique solution.

Minor Comments

The state of shallow megathrust coupling is an important issue in subduction zone earthquake science, which the authors explicitly recognize in the Introduction (pp. 2-3). However, as far as I can tell, they do not return to the issue of megathrust coupling in their Discussion. If their analysis cannot reveal anything about the shallow coupling on the megathrust, that is fine, but then I would prefer they talk less about it in the Introduction.

It has been proposed that variations in shallow coupling are influential on the occurrence of the primary fast-faulting strike slip rupture, so it is an important context, but we also believe that gradients in shallow coupling are likely involved in the occurrence of the upper wedge lateral compression, and we expand on this in the discussion. Both points indeed remain largely conjectural, as we describe in the revised summary, given that shallow coupling is actually not resolved geodetically, and the relatively deep upper boundary of the slip zones for the 2020 Simeonof and 2021 Chignik events (both around 25 km deep) leave open the question of coupling beneath the continental slope. We discuss this issue in the revised Discussion.

One of the most important sections is about the poor fit to tsunami using the fast-slip model that fits the seismic and geodetic observations (p. 5, ll. 15ff.). I understand that Nature Communications is a short format journal, but in my opinion this section is too important to be given such little space. I would expand it with more detail if possible.

We add specific discussion of the inadequacy of the tsunami waveform fitting for the fast-slip faulting in our discussion of Figure 4, as suggested.

Detailed Comments

p. 1, l. 5: The phrasing in the first sentence of the Abstract is unclear and sets a vague framework for the context of this study. Were slow, shallow ruptures not considered

hazardous prior to this paper? I might just remove this sentence and start with the next sentence.

We modify the sentence to clarify that the issue about slow tsunamigenic ruptures in the shallow megathrust and wedge is a pressing concern (certainly it predates this paper, but this paper adds to that concern).

p. 1, ll. 13-14: I think it is important to indicate the uncertainty of the slow, tsunamigenic source somewhere in the Abstract, indicating that it is a plausible scenario consistent with the datasets investigated here.

We add caveats to convey the overall uncertainty of the slow slip component in the revised abstract (word count is very limited) and more discussion in the main text, as requested.

p. 2, l. 14: It might be useful to provide a sense of hypocenter depth uncertainty. I suspect that most of these events could have occurred within the Pacific slab.

We add discussion of the non-uniqueness of the fast-slip secondary source, and present in the Supplement results for alternate, adequately fitting models with the secondary rupture in the wedge and in the slab.

p. 2, ll. 14-16: I would add that most large, intraslab ruptures are under or seaward of the trench, not 50+ km arcward of the trench

We add this fact in the revision, as suggested.

p. 4, l. 8: How much worse is the fit when slip is confined to larger depths?

As now described more extensively in the revision, the seismo-geodetic fitting is not very diagnostic of depth or geometry, and we present alternate adequately fitting models in the Supplement for the reader to examine. However, the plausibility of the shallow north-dipping faulting is enhanced due to the known existence of a candidate fault structure, as noted by the reviewer, and the unreasonable nature of the intraslab model in terms of transecting the strike-slip faulting and having slip at opposite ends of the slip model, which we deem to be unlikely.

p. 4, ll. 11ff: In this paragraph, I think that statements like “fitting the teleseismic data better,” “concentrates slip within the fault model,” “a more stable and sensible solution,” and “typically fast” are too qualitative

We revise the model fitting descriptions to be as quantitative as is justified. The language is still intrinsically informal given that model uncertainty is large and cannot be fully quantified due to nonlinearity.

p. 5, ll. 12-13: Although potentially true, this last sentence is too speculative to be

useful. I would remove it or run a quick back-slip model with a transition in coupling (easy enough to do with a similar setup as the Coulomb stress modeling) and resolve the stresses on the second fast-slip component to see if slip on that structure is favored. That way, the authors could be justified in making the assertion that this specific faulting is favored by a transition in coupling.

The cited reference performed calculations for assumed laterally varying coupling models to account for the intraslab shear geometry, and we do not replicate those published results. The constraint on coupling is, in our opinion, actually too limited to justify additional modeling here. We clarify that statement.

p. 6, ll. 17-20: One thing that might help me in this discussion is to label one or more of the tsunami waveforms in Fig. 4 with the impulsive peak and the wide trough. That might help make this result more concrete. In general, I think that the authors could leverage their nice figures even more in the text.

The discussion of the waveform features and the misfit of the fast-slip contributions is expanded to specifically focus the reader on the key observations. Labeling the figure with the clear features is not needed in our opinion.

p. 7, l. 8: How large is the “large time discrepancy?”

We specify the arrival time offset in the revision, as suggested. It was specified previously in discussion of Fig. 4.

p. 7, l. 20: “to account for run-out effects while conserving volume” should have a reference and/or explanation

We now cite the work by Okal and Hebert again, which developed the model.

p. 8, ll. 3-6: These first two sentences of this paragraph seem like they should come at the beginning of the section.

We mention that this point was noted previously; here it is restated to focus on timing of the second arrival which was not accounted for by the fast-slip model.

p. 8, ll. 7ff: There are some vague or qualitative statements in here that could be made more precisely. What is the azimuth covered by the DART buoys? Is dipole orientation also a factor in tsunami waveform phase, or mostly location? What does it mean for seafloor motion to be “sensitively determined by finding consistent and accurate arrival time at each DART?” What were the parameter ranges in the search? What tradeoffs are there in the parameters?

We point out the DART buoy distribution shown in Figure 2 to clarify the nearly 180° azimuth range covered by those stations. We clarify the language associated with the DART distribution and waveform fitting, adding discussion of the tradeoffs.

We move up a later sentence that describes the location and amplitude sensitivity of the dipole.

p. 9, l. 12: I think the authors could more explicitly set up why they prefer faulting to a landslide/slump source

We point out that the simplified dipole model is not actually a sensible physical model, as it cannot predict the horizontal motions that will likely violate the hr-GNSS observations (as shown for the physical models of megathrust and splay slow faulting).

pp. 9-10: I think these sections could be merged without a significant loss of clarity

We keep the sections separate, to help the reader follow the progression toward a final model.

p. 11, l. 1: It seems that a statement is needed to introduce the subsequent discussion such as, "Assuming this unusual, unobserved, slow faulting is the source of the tsunami..."

We add qualification of the discussion as suggested.

p. 11, ll. 3-4: This is another speculative statement. Remove these or support them with evidence.

We note that slow slip in weak sediments may be involved.

p. 13, l. 14 and l. 19: Maybe it is convention for the data types, but it seems strange to me to report frequency and period bands right next to each other

We use period ranges consistently in the revision, as suggested.

p. 14, ll. 5-6: Are the faults constrained to have no obliquity at all? Also, "right lateral strike slip" and "normal slip" are not strictly speaking "rake angles."

We clarify the two rake end-members used purely strike-slip and purely dip-slip.

p. 14, ll. 11-12: Equal weighting can take some different forms, so I would prefer to see an equation for calculating misfit

We clarify that the weighting is equal for data-functionals for the GNSS statics and seismic waveforms. This is a standard procedure for combining vector and time series data.

p. 16, ll. 22ff: I am not fond of the repetition in this section from the previous section. At first I thought the authors had made a mistake in their writing.

We remove the redundant description of the modeling, as suggested.

Fig. 1: I think the cross-section should have the same orientation as in Fig. 10.

Figure 10 is from a north view, while Figure 1 is from a south view. We prefer the figures as they are; we think there is no confusion in orientation.

Fig. 3: The image should use the full range of the color scale, unless using exactly the same color scale in another figure (which is not the case, because the scale in Fig. 7 has a different range). Also, please modify the figures to use color-blind friendly color maps (not rainbow, jet, lots of reds/greens, etc.)

We change the scale of the deformation plot to match the peak-to-peak range of the calculated values, as suggested. The scales in Figure 7 vary to help show the results for the separate fast-slip, slow-slip, and combined models clearly.

Fig. 7: This is a very complicated paneled figure, and might need to be broken into parts. It is so dense that I thought some of the panel labels referred to other adjacent panels. I think the GNSS timeseries should zoom into the 0-100 second range and ignore the (mostly zero) time series from 100 to 600 seconds.

We add panel labels to the figure to make it clear for the reader. The reviewer appears to miss the point of the long time series comparison, which is to show that the slow-slip process, which ruptures from 30 to 330 s does not produce unacceptable seismic and static ground motions at the stations shown. This is stated more clearly in the revision when discussing Fig. 9. Similar figures in the Supplement demonstrate that alternate models do produce unacceptable ground motions over the same long time interval.

Fig. 9: This does not seem to be necessary as a primary figure, although I do like it (with a better color map).

We now emphasize that Figure 9 relates back to the first line of the abstract. Slow slip process up-dip of the deeper large earthquake in 2020 produced much larger tsunami. Discussion of this is added in the assessment of the figure.

We thank the reviewer for the thoughtful comments and acknowledge their contribution in the revision.

Reviewer #2 (Remarks to the Author):

Bai and colleagues present a combined analysis of GNSS, seismic and tsunami data to understand an enigmatic M7.6 earthquake that occurred in the Alaska subduction zone in October 2020. Unlike other recent, large earthquakes in this region, this event was an intraplate event. The authors convincingly show that slip on structures

in the overriding and subducting plates is required to explain all of the observations, including a secondary tsunami source. This paper is significant because it offers the opportunity to investigate complex interactions between faults in the subducting and overriding plates and their implications for seismic and tsunami hazards. Overall, I found the paper well written and illustrated. I am not an expert in the types of analyses presented, but I generally found the analyses clearly described and convincing. I have a few comments and questions that I think can be mostly addressed with minor revisions to the manuscript.

The authors require slip on two faults in the overriding plate to explain their observations. One of my primary comments is that I think the paper would be much stronger if the authors considered other geological/geophysical evidence for faulting in the overriding plate to support and contextualize their observations. I support the author's approach of first guiding their definition of ideal fault geometries and slip based on the analysis of the seismic, geodetic and tsunami data, alone. However, given the understandable uncertainty in solutions, the manuscript would be stronger if then provided geological and geophysical data to support their results and potentially guide the choice of preferred solutions. I provide specific comments/suggestions on this and other points below.

-The authors consider two geometries for a fault in the upper plate that they require to explain the non-double couple component of the M7.6 earthquake: one is south dipping and the other north dipping. The authors prefer the south dipping fault because of a better fit to the data (pg 4, Line 19). However, a north dipping structure might be more geologically plausible based on the fact that the upper plate of this subduction zone comprises a series of accreted terranes separated by north dipping sutures (Horowitz et al., 1989; Rowe et al., 2011) and that a major north dipping fault has been imaged in seismic reflection profiles (Bécel et al., 2017, von Huene et al., 2019), which may have reactivated one of these collisional sutures (Shillington et al., 2022). The difference in data fit between the north and south dipping upper plate fault is not quantified in the paper. How much better is the fit for the south dipping fault than the north dipping fault, and is it significant in light of uncertainties? Given that a south dipping fault is less consistent with other existing observations, I think the paper needs to make a stronger case for preferring this geometry. If the difference in misfit is not significant, I think that a north-dipping fault is more consistent with what we know about the upper plate of this margin.

We very much appreciate the reviewer's discussion of the choice of second fast-fault geometry, and conducted extensive additional modeling, exploring the depth, geometry, and orientation of the faulting. This further model exploration established multiple geometries that can fit the seismo-geodetic data comparably well. North-dipping and south-dipping faults in the upper plate and in the slab are all viable given the waveform fitting. This is largely because the associated faulting is much smaller than the primary strike-slip faulting (10% of the seismic moment) and in all models it occurs late in the process, contributing to later P and S wave arrivals.

Given the lack of specific resolution from waveform data, we now invoke the a priori information on the known existence of a candidate fault as imaged in the seismic reflection profiles as a basis for a preferred north-dipping faulting in the upper plate, as suggested by the reviewer. We prefer this over the intraslab possibilities based on the finding that successful models involving a north- or south-dipping geometry transect the intraslab strike-slip fault, with the models having slip distributed on each end of the fault, which is usually an indication of instability in the inversion. The original shallow south-dipping geometry is viable in terms of waveform fitting, but lacks any a priori known structure. We clarify the basis for our final preference in the revision, and include in the supplement two models for alternate cases (shallow south-dipping and intraslab north-dipping) cases so the reader can see the slip models and some of the waveform fits for those possibilities (we do not include the many pages of total waveform fits from each joint inversion, but the fits are very similar to those for the preferred model, which are shown in the Supplement).

-If a north dipping fault is used to explain the fast rupture component of this event, what are the implications for the location and geometry of the secondary tsunami source?

The slow rupture component proves very insensitive to the choice of secondary faulting for the fast-rupture component because the latter is barely tsunamigenic. We now include subsets of waveform fits and all tsunami fits for the north-dipping intraslab and shallow south-dipping cases in new Supplementary Figs. 12-15 to document the stability of the slow-slip model relative to the uncertainty in the fast-slip model.

-The section entitled “Constraining the secondary tsunami source” could be written more clearly and succinctly. I found some of the text repetitive (e.g., pg 8, Line 3-8). On the other hand, some of the logic in this section was not clear to me as written. If I understand, the dipole modeling approach used to narrow down possible locations for the secondary source could apply to either a shallowly dipping fault or to a submarine land slide. However, the authors appear to rule out a landslide as a possible secondary tsunami source and only consider slow-faulting scenarios in the rest of the paper. This choice and supporting logic should be laid out more explicitly (e.g., by rewriting and expanding Lines 7-11, pg 9, if that is the rationale).

The reviewer’s interpretation is correct, and we edit the text to be more succinct and remove any redundant description. The dipole model has been used in prior work as a first-order representation of slumping, but it is not a reasonable model for evaluating seismic excitation (essentially, one can represent a slump with a shallow normal-faulting geometry, but the key constraint is that the slump must go downslope and this will result in violating the nearby hr-GNSS observations in the same way that slow megathrust and slow splay faulting are documented to do). Rotating the dipole (or normal-faulting) to reduce the deformation at the hr-GNSS stations violates the down-slope slumping requirement, reducing the viability of any slump interpretation. We explain this more clearly in the revision.

-The interpretation of a trench-normal upper plate thrust fault experiencing slow slip is very interesting, and its quite remarkable if such a fault would produce no seismic or geodetic signals. The paper would be stronger if it explored other evidence for this fault (e.g., Lines 5-10, Pg 12). Is there any other possible dataset that could have detected such a slow slip event (e.g., InSAR given the proximity to the Shumagin Islands)? Is there any other evidence of trench normal compression (e.g., from earthquake focal mechanisms, etc)? Or other geological support for such a fault? This region is near (although east of) the edge of the Beringian margin, where trench oblique structures are observed – could the orientation of this structure plausibly parallel those? I think the paper would be stronger if there was more discussion about this feature since it is a major conclusion of the paper and a very surprising one. It is not very satisfying that the paper proposes this extraordinary fault, but does not discuss it in much detail.

The computed static deformations are negligible at the nearby GNSS stations, so InSAR across the more remote Shumagin Islands will not observe the proposed slow-slip faulting, as we now note in the revision. Earthquake focal mechanisms are very few in number near the continental slope, and we now show in new Supplementary Figure 20 the variable mechanisms for several of the larger shallow aftershocks, which have various oblique compression and normal faulting mechanisms, but none match the geometry of the slow-slip source. However, the slow slip process is intrinsically consistent with a total absence of seismic activity on the corresponding fault, so this is really a neutral result. We now note the Beringian margin to the west, but do not have knowledge of any additional data that can test for this structure. We do not know of any additional information that is available at this time with which to further test the model geometry.

Other minor comments:

-Pg 7, Lines 13-15: I am not familiar with this type of parameterization, so the description of a trough, ridge and level arm was not clear to me. Suggest rephrasing or elaborating a bit more (or adding a figure to the supplement to illustrate).

The description of the seafloor deformation for the dipole model is clarified in the revision.

-Figure 10 labels the updip portion of the megathrust as weakly coupled even though the text correctly states that coupling on the offshore, shallow megathrust is poorly known (pg 2, Lines 23-25). Is a downdip change in coupling important to the results presented here?

We add a question mark to Figure 10 labeling of up-dip coupling, which is taken from work by others. Prior papers, mostly based on geodetic observations, have suggested the coupling is weak along the Shumagin segment and strong along the Semidi segment, but the near-trench coupling is really not resolved by those data, as we state. The distribution of downdip coupling is also not well-resolved, as we note;

it can successfully be restricted to the areas that failed coseismically in 2020, which likely has little affect on the region of the strike-slip event. As shown in Fig. 1, our preferred models for the 2020 Simeonof and 2021 Chignik events have very similar along-dip coseismic slip distributions and we suspect the same is true for the 1938 event. It is not clear that there is much change in downdip coupling that could control the stress state in the shallow wedge.

Thank you for the opportunity to review this interesting paper,
Donna Shillington

We thank Professor Shillington for the thoughtful comments and acknowledge her contribution in the revision.

Reviewer #3 (Remarks to the Author):

(Jeff Freymueller)

This paper addresses a lingering mystery of the October 2020 Sand Point earthquake: why this strike-slip (or mainly strike-slip) earthquake that occurred primarily in the downgoing oceanic plate generated a larger tsunami than the larger, megathrust July 2020 Simeonof earthquake that preceded it.

The authors first develop a model, which they refer to as the “fast slip” model, based on seismic waveforms and GNSS displacements. Their model is more complex than previously published models, and includes fault slip both above and below the plate interface. This model is well described, and fits the data used to derive it but does not explain the resultant tsunami. (Nor has anyone else’s model fit the tsunami). This leads them to consider a secondary tsunami source, which needs to start 4-5 minutes after the mainshock to explain the DART buoy record. The challenge here is that one key constraint on this secondary seismic source is that it must not produce any seismic or geodetic signature. As a result, most straightforward fault models don’t work because they would produce non-zero displacements at the GPS sites.

This is a very good summary of the challenge presented by this event.

The authors finally propose that the secondary tsunami source was slow rupture on a thrust fault on the continental shelf that strikes nearly trench normal. The magnitude of slip is very high (17 meters), and the resulting seismic moment is nearly as large as the “fast slip” earthquake model itself. The source is oriented so that the main seafloor displacement occurs in the right place, and that there are no displacements observed at the GPS sites. Lower slip on a larger fault might fit the tsunami data, but it would be harder for such a fault to remain invisible to the geodetic data and the authors note that it might not explain the tsunami data as well. The rupture was presumed to be slow enough to be invisible to seismometers. Is it really true that a ~300 sec slow rupture of $M \sim 7.5$ would leave no seismic signature at all? Might there not be some long-period signal associated with that? Substantially smaller events resulting from glacial sources have in fact been

detected. However, I could certainly be wrong about the source duration of those events.

In the revision, we explored a broad suite of models for the slow slip faulting with the fast-faulting updated to have a north-dipping secondary event. We explored various lengths, widths, depths and positions of the slow-slip component, finding slip values that best-fit the tsunami data without violating the hr-GNSS observations. As discussed, our preferred model, in terms of fitting the DART and tide gauge tsunami data best, involves a 20 km x 20km fault with 15 m slip, but models with a 30 km x 30 km fault with 7 m slip or 40 km x 40 km with 4 m slip do quite well. We include results for the latter two models in the Supplement (new Supplementary Fig. 16, 17) to inform the interested reader. The tsunami fitting for the preferred model in the main text is systematically, but only slightly, better than for the larger rupture area, lower slip cases, and we do slightly prefer the smaller rupture area, larger slip case. We convey the model uncertainty better in the revision. We show that this range of models does not violate the nearby hr-GNSS observations.

The long-period excitation for the slow slip component is explicitly shown in Supplementary Fig. 6, with the amplitudes being much weaker than the strong radiation from the fast-slip component. Added to the lack of detectability is that the phase is strongly shifted by the delayed onset time and 5-minute duration of the slow slip radiation, so the amplitude spectra do not add linearly. The detectability of slow slip radiation in the coda of larger moment fast slip is intrinsically problematic, there is no clear evidence in our large seismic and geodetic data sets that for periods shorter than 250 s that indicates the slow-slip process. Longer period signals are not well-excited and must involve complex interferences among the fast- and slow-slip processes, so it would be very hard to resolve for this compound event.

As far as I can tell, the model does fit the data constraints. I still find it hard to believe, though. It is intrinsically hard to dispel all skepticism about a component of a model source when one of the primary constraints is that it must be invisible to the data sets used to develop the slip distribution. I also have to wonder why such a fault would exist in the first place – is there any evidence for it? – and what would make it able to slip such an extraordinary amount on a relatively small fault patch? What would load a fault of this orientation to bring it close to failure? As a result, this model seems to me to be highly speculative.

The reviewer's skepticism is reasonable; we only came to the preferred model after ruling out the more obvious possibilities, such as slow slip on the megathrust, slow slip on a splay fault, slumping, etc. We agree the model has non-uniqueness and we do convey that better in the revision, but the tsunami generation is unambiguous and gives relatively tight constraints on the location and amount of seafloor motion that occurred. This leads to a specific fault model that accounts for the required seafloor motion, but the cause of the implied along-trench compression in the shallow wedge is not apparent. We do not speculate much on the loading process, as it remains unclear, and targeted imaging of the source region with reflection

seismology and high resolution bathymetric scanning is the most promising approach to testing the model, as we emphasize in the revision.

What I would like to see from the authors is more support for how this is the only possible explanation of the tsunami. Unless that is demonstrated, then all they can really do is put this forward as a possible explanation that doesn't violate any data we now have (there is also a GPS-Acoustic seafloor displacement that will become available, but it is not yet published). Here are my key remaining questions:

We clearly state that this is a demonstration of existence of a viable model, not an assertion of uniqueness. We rule out more conventional possible explanations; but we cannot rule out all possible unconventional explanations. But lest this seems too speculative, one need only consider the differences in Figure 4 (which shows that the fast-slip model cannot match the tsunami data at all) with Figure 8 (which shows that our preferred model does a remarkable job of fitting the data recorded in Alaska and Hawaii). The information contained in tsunami waveforms for simple deep water paths is very high, so such good waveform fitting provides good space-time constraint on the seafloor deformation provided by our preferred model. The specific faulting models that produce good waveform fitting without violating any other observations represent a viable class of models; we note that several unpublished studies have failed to achieve this.

Would such a large event with ~300 sec duration actually be invisible to long-period seismology? Can you back up such an assertion more thoroughly?

We discuss the long-period spectral amplitudes shown in Supplementary Figure 6 more in the text.

Is there any evidence in past seismicity, bathymetry, or mapped faults for the structure that they have proposed?

The seismic reflection profiles do not go directly through the candidate fault area, so they cannot support or exclude the model geometry (and they would be insensitive to orthogonal dipping structure. Available bathymetry maps are not high resolution right in the source area, but no clear feature is apparent, as we now note.

Why would a structure be tectonically loaded, given that it is nearly orthogonal to plate convergence?

Large deformation in weak accretionary toes can occur without seismicity, but usually involves trench-parallel faulting. The oblique geometry here may have been loaded by prior strike-slip faulting in the subducting plate interacting with the lateral gradient in western structure of the accreted terranes as is now discussed, or possibly by the almost unconstrained plate boundary coupling beneath the continental slope. We do not invoke a specific origin for the deformation as it is

highly speculative, but the seafloor deformation needed to account for the tsunami excitation is well constrained.

Large slip has been observed in tsunami earthquakes on the very shallow megathrust, but this proposed source is not on the megathrust; it would cut what we presume to be normal forearc upper crust. Can they point to any other examples of very large slow slip on a fault in a non-megathrust setting?

The 2016 Kaikoura strike-slip event in South Island New Zealand included slow thrust faulting of comparable slip (9 m) on distinct thrust faults offset from the main strike-slip faults, but we feel it would confuse the reader to discuss this unusual observation. We do include more references to examples of slow deformation in the shallow megathrust, as those are more clearly relevant.

Overall, the writing and organization of the paper is very good, so there are almost no minor corrections. Just one:

Page 9, line 24. Note that a long source process may obscure the seismic expression, but it will not obscure the (static) geodetic expression of the deformation due to the slip. Also, as noted above, I would like to see a stronger argument that such a large moment event would remain invisible at long periods (it may be that it would, but the assertion should be backed up given that long-period sources of magnitude much smaller than this have been detected and reported).

In the revision we emphasize that by including the 600 s hr-GNSS signal calculations we demonstrate that the dynamic AND static deformations for our model are not detected at the most sensitive, nearby stations. The dynamic signals (essentially near-source Love and Rayleigh wave energy) are actually the hardest to make consistent with the hr-GNSS; even when the static terms are negligible, dynamic motions from long period surface waves are present and these rule out the megathrust, splay and slump models due to their focal mechanisms.

We thank Professor Freymueller for the thoughtful comments and acknowledge his contribution in the revision.

REVIEWERS' COMMENTS

Reviewer #1 (Remarks to the Author):

Bai et al. have thoroughly dealt with all of the issues I identified in my previous review. I appreciate their responsiveness, and I think their current paper is a strong contribution to our understanding of this strange earthquake.

One small note: in l. 314, the authors state that the larger shallow aftershocks show a variety of thrust and extensional mechanisms. My recollection is that the aftershocks mostly have strike-slip and normal faulting mechanisms in this region, consistent with the authors' parameterization of the two fast-slip segments.

Reviewer #2 (Remarks to the Author):

Bai and colleagues present a combined analysis of GNSS, seismic and tsunami data to understand an enigmatic M7.6 earthquake that occurred in the Alaska subduction zone in October 2020. The authors convincingly show that slip on structures in the overriding and subducting plates is required to explain all of the observations, including a secondary tsunami source. I reviewed an earlier version of this manuscript, and I find the revision is much improved and addresses nearly all of my comments on the previous version. I just have a few more very minor suggestions.

-I found the explanation given for ruling out a slope failure for the secondary tsunami source in the rebuttal letter very clear and convincing, but I do not think the text of the manuscript is clear on this point even in the revision. I encourage the authors to add some sentences to more explicitly explain their rationale/evidence for excluding the slope failure explanation.

-I think the paper would have a stronger ending and be more impactful if the authors expanded the last sentence into a paragraph on the significance of the interpreted fast and slow slip on structures in the overriding and subducting plates for deformation and hazards in subduction zones generally.

Other minor points:

-Line 40 – consider changing to “parallel to the trench”

-Figure 1 – consider adding an inset to Figure 1a to show the location of this area with respect to Alaska-Aleutian trench

-My name is misspelled in the acknowledgements :).

Donna Shillington

Reviewer #3 (Remarks to the Author):

I have read over the response letter and the changes to the manuscript, and I agree that the authors have done a better job in acknowledging the non-uniqueness and possible issues with their proposed slow slip source. So I think they have been responsive to the review comments. Although I focused on the parts of the manuscript where the changes were made, I did notice a couple of other things, both minor wording issues.

Page 7, line 11-12. Something about this sentence is not quite right: “... show no deformation after the first 60 s that are well accounted for by the two-fault model.” I suspect an editing error, as the parts before and after “60 s” seem like they should be part of two separate statements. The data do indeed show no deformation after the first 60s, and the deformation that is observed BEFORE 60s is well accounted for by the model.

Page 11, line 19. This needs rephrasing or may be missing a word: " the need to have drawn down on the continental shelf". The water would have been drawn down on the continental shelf to produce the tsunami, but the text as written makes it unclear whether they meant "have draw down" or something else.

Response to 2nd reviews of "Fast and slow intraplate ruptures during the 19 October 2020 magnitude 7.6 Shumagin earthquake" by Bai et al. The reviewer comments are reproduced below in black type, and our responses and indications of how we have revised the manuscript to address the comments are shown in blue.

REVIEWER COMMENTS

Reviewer #1 (Remarks to the Author):

Bai et al. have thoroughly dealt with all of the issues I identified in my previous review. I appreciate their responsiveness, and I think their current paper is a strong contribution to our understanding of this strange earthquake.

One small note: in l. 314, the authors state that the larger shallow aftershocks show a variety of thrust and extensional mechanisms. My recollection is that the aftershocks mostly have strike-slip and normal faulting mechanisms in this region, consistent with the authors' parameterization of the two fast-slip segments.

We clarify our description to say "oblique strike-slip and normal" faulting. Figure S20 shows the moment tensor solutions of several shallow aftershocks in the GCMT catalog.

We appreciate the reviewer's constructive suggestions in finalizing the manuscript.

Reviewer #2 (Remarks to the Author):

Bai and colleagues present a combined analysis of GNSS, seismic and tsunami data to understand an enigmatic M7.6 earthquake that occurred in the Alaska subduction zone in October 2020. The authors convincingly show that slip on structures in the overriding and subducting plates is required to explain all of the observations, including a secondary tsunami source. I reviewed an earlier version of this manuscript, and I find the revision is much improved and addresses nearly all of my comments on the previous version. I just have a few more very minor suggestions.

-I found the explanation given for ruling out a slope failure for the secondary tsunami source in the rebuttal letter very clear and convincing, but I do not think the text of the manuscript is clear on this point even in the revision. I encourage the authors to add some sentences to more explicitly explain their rationale/evidence for excluding the slope failure explanation.

We augment the discussion of why we do not favor slumping in the text, as suggested.

-I think the paper would have a stronger ending and be more impactful if the authors expanded the last sentence into a paragraph on the significance of the

interpreted fast and slow slip on structures in the overriding and subducting plates for deformation and hazards in subduction zones generally.

We follow this suggestion and expand the conclusion accordingly.

Other minor points:

-Line 40 – consider changing to “parallel to the trench”

We make the suggested change.

-Figure 1 – consider adding an inset to Figure 1a to show the location of this area with respect to Alaska-Aleutian trench

We considered this, but think that Figure 2 provides a sufficient big picture context. So we prefer not to add further complexity to Figure 1.

-My name is misspelled in the acknowledgements :). Donna Shillington

We sincerely apologize for misspelling the reviewer’s name, and we have corrected the mistake.

We thank Professor Shillington for the insightful perspective and valuable input.

Reviewer #3 (Remarks to the Author):

I have read over the response letter and the changes to the manuscript, and I agree that the authors have done a better job in acknowledging the non-uniqueness and possible issues with their proposed slow slip source. So I think they have been responsive to the review comments. Although I focused on the parts of the manuscript where the changes were made, I did notice a couple of other things, both minor wording issues.

Page 7, line 11-12. Something about this sentence is not quite right: “... show no deformation after the first 60 s that are well accounted for by the two-fault model.” I suspect an editing error, as the parts before and after “60 s” seem like they should be part of two separate statements. The data do indeed show no deformation after the first 60s, and the deformation that is observed BEFORE 60s is well accounted for by the model.

We clarify this by breaking the sentence into two sentences.

Page 11, line 19. This needs rephrasing or may be missing a word: “the need to have drawn down on the continental shelf”. The water would have been drawn down on the continental shelf to produce the tsunami, but the text as written makes it unclear whether they meant “have draw down” or something else.

We clarify the phrasing. The seafloor drawdown that produces the trough in the tsunami signal must locate at the inland of the continental shelf break; if too far inland the wave will be trapped by the shelf.

We thank Professor Freymueller for the careful examination and thoughtful comments.